# Spectral Informed Neural Network

## Abstract

In scientific computing, the utilization of physics-informed neural networks (PINNs) for solving partial differential equations (PDEs) has been burgeoning. More accurate and efficient PINNs are required and under research. One bottleneck of current PINNs is the computation of the high-order derivatives via automatic differentiation which often necessitates substantial computing resources especially when dealing with complex PDEs and high-dimensional problems. To tackle this, we propose a spectral-based neural network that substitutes the differential operator with a multiplication. Compared to PINNs, our approach requires less GPU memory and a shorter training time. Furthermore, thanks to the exponential convergence of the spectral basis, our approach is more accurate. Moreover, to handle the different situations between the physics domain and the spectral domain, we provide a strategy to train networks using their spectral information. Through a series of comprehensive experiments, we validate the aforementioned merits of our proposed network.

## 1. Introduction

With the rapid advancements in machine learning and its related theories, integrating mathematical models with neural networks provides a novel framework for scientific research. The representative methods are the Physics-Informed Neural Networks (PINNs) (Lagaris et al., 1998; Raissi et al., 2019) and the Deep Ritz method (Yu et al., 2018). Thanks to the development of the Monte Carlo method (Rubinstein & Kroese, 2016), the automatic differentiation (AD) (Baydin et al., 2018), and the universal approximation theorem (Hornik, 1991), PINNs have garnered significant attention because of their ability to solve partial differential equations (PDEs) without suffering from the curse of dimensionality (CoD) (Wojtowytsch & Weinan, 2020; Han et al., 2018), compared with traditional numerical methods such as Finite Difference Methods (FDM), Finite Element Methods (FEM), Finite Volume Methods (FVM). Moreover, PINNs also demonstrate the merits in handling im-

perfect data (Karniadakis et al., 2021), extrapolation (Yang et al., 2021; Ren et al., 2022), and interpolation (Sliwinski & Rigas, 2023). These capabilities have propelled PINNs into wide applications, including but not limited to fluid dynamics (Jin et al., 2021), aerodynamics (Mao et al., 2020), surface physics (Fang & Zhan, 2019), power systems (Misyris et al., 2020), and heat transfer (Gao et al., 2021).

Because of the requirements of derivatives with respect to the inputs of networks, using AD to compute the gradient is inefficient and computationally expensive in PINNs. Furthermore, (hoon Song et al., 2024) provides a complete theorem stating that: to keep the same convergence rate, the required width of the network grows exponentially as the PDE order $p$ increases, which reveals that vanilla PINNs face challenges in learning high-order PDEs. To address this problem, various approaches have been proposed that leverage alternative numerical methods to replace AD. PhyCR-Net (Ren et al., 2022; Rao et al., 2023) utilizes the FDM to replace AD; DTPINN (Sharma & Shankar, 2022) applies the finite difference for radial basis functions instead of computing high-order derivatives; sPINN (Xia et al., 2023) employs the spectral method of orthogonal polynomials to avoid computing the derivatives. Additionally, SVPINN (Lyu et al., 2022; hoon Song et al., 2024) splits high-order derivatives into several low-order derivatives. Similarly, DFVM (Cen & Zou, 2024) adopts this approach but calculates these low-order derivatives by the FVM rather than AD.

On the other hand, scholars have been devoting their attention to improving the efficiency of AD for a long time (Bendtsen & Stauning, 1997; Karczmarczuk, 1998; Wang, 2017; Laurel et al., 2022). The representative methods are the Taylor-mode (Griewank & Walther, 2008) for univariate derivatives and Stochastic Taylor Derivative Estimator (Shi et al., 2024) for multivariate derivatives. Researchers (Wang et al., 2022a; Shi et al., 2024; Hu et al., 2024) have already utilized this approach within the framework of PINNs to obtain high-order derivatives in Poisson's equations and Kuramoto–Sivashinsky equations.

In this paper, we propose Spectral-Informed Neural Networks (SINNs) as an efficient and low-memory approach for training neural networks to solve partial differential equations (PDEs) by making use of the spectral information derived from the spectral domain. When compared with

Physics-Informed Neural Networks (PINNs), our SINNs utilize a precise and efficient alternative to automatic differentiation (AD) to compute spatial derivatives. The input is the frequencies of the spectral basis instead of the grid points from the physical domain, and the output is the coefficients in the spectral domain rather than the physical solution. Moreover, the property of exponential convergence in spectral methods when approximating any smooth function (Canuto et al., 1988; Orszag, 1971) enables SINNs to achieve higher accuracy.

Our specific contributions can be summarized as follows:

- We propose a method that eliminates the spatial derivatives of the network to deal with the high GPU memory consumption of PINNs.

- We propose a strategy to approximate the primary features in the spectral domain by learning the low-frequency preferentially to handle the difference between SINNs and PINNs.

- We provide an error convergence analysis to show that SINNs are more accurate than PINNs. Furthermore, our experiments corroborate that the method can reduce the training time and improve the accuracy simultaneously.

The paper is structured as follows: In Section 2, we provide a concise overview of PINNs and discuss AD and its developments. Using a simple experiment, we highlight the challenge encountered in computing high-order derivatives within PINNs. To address this drawback, in Section 3, we propose our SINNs and give an intuitive understanding by a concrete equation. In Section 4, we demonstrate state-of-the-art results across a comprehensive experiment. Finally, Section 5 provides a summary of our main research and touches upon remaining limitations and directions for future research.

## 2. Physics-informed neural networks (PINNs)

We briefly review the physics-informed neural networks (PINNs) (Raissi et al., 2019) in the context of inferring the solutions of PDEs. Generally, we consider PDEs for $\boldsymbol{u}$ taking the form

$$
\begin{aligned}
\partial_t \boldsymbol{u} + \mathcal{N}[\boldsymbol{u}] &= 0, \quad t \in [0, T], \ \boldsymbol{x} \in \Omega, \\
\boldsymbol{u}(0, \boldsymbol{x}) &= \boldsymbol{g}(\boldsymbol{x}), \quad \boldsymbol{x} \in \Omega, \\
\mathcal{B}[\boldsymbol{u}] &= 0, \quad t \in [0, T], \ \boldsymbol{x} \in \partial\Omega,
\end{aligned} \tag{1}
$$

where $\mathcal{N}$ is the differential operator, $\Omega$ is the domain of grid points, and $\mathcal{B}$ is the boundary operator.

The ambition of PINNs is to obtain the unknown solution $\boldsymbol{u}$ to the PDE system (1), by a neural network $\boldsymbol{u}^\theta$, where $\theta$ denotes the parameters of the neural network. The constructed

loss function is:

$$
\mathcal{L}(\theta) = \mathcal{L}_{ic}(\theta) + \mathcal{L}_{bc}(\theta) + \mathcal{L}_r(\theta), \tag{2}
$$

where

$$
\begin{aligned}
\mathcal{L}_r(\theta) &= \frac{1}{N_r} \sum_{i=1}^{N_r} \left| \partial_t \boldsymbol{u}^\theta \left( t_r^i, \boldsymbol{x}_r^i \right) + \mathcal{N} \left[ \boldsymbol{u}^\theta \right] \left( t_r^i, \boldsymbol{x}_r^i \right) \right|^2, \\
\mathcal{L}_{ic}(\theta) &= \frac{1}{N_{ic}} \sum_{i=1}^{N_{ic}} \left| \boldsymbol{u}^\theta \left( 0, \boldsymbol{x}_{ic}^i \right) - \boldsymbol{g} \left( \boldsymbol{x}_{ic}^i \right) \right|^2, \\
\mathcal{L}_{bc}(\theta) &= \frac{1}{N_{bc}} \sum_{i=1}^{N_{bc}} \left| \mathcal{B} \left[ \boldsymbol{u}^\theta \right] \left( t_{bc}^i, \boldsymbol{x}_{bc}^i \right) \right|^2,
\end{aligned} \tag{3}
$$

corresponds to the three equations in (1) individually; $\boldsymbol{x}_{ic}^i, \boldsymbol{x}_{bc}^i, \boldsymbol{x}_r^i$ are the sampled points from initial constraint, the boundary constraint, and the residual constraint; $N_{ic}, N_{bc}, N_r$ are the total number of sampled points correspondingly.

### 2.1. Automatic differentiation (AD)

AD gives the required derivative of an overall function by combining the derivatives of the constituent operations through the chain rule based on evaluation traces. Herein, in PINNs, AD is also used to calculate the derivatives with respect to the input points. However, AD demands both memory and computation that scale exponentially with the order of derivatives by the scaling $\mathcal{O}(d^p)$ where $p$ is the differentiation order and $d$ is the dimensionality, although there are investigations (Griewank & Walther, 2008; Bettencourt et al., 2019; Tan, 2023) on computing high-order derivatives efficiently by Faà di Bruno's formula[1] and Taylor mode[2]. Alternatively, by replacing the high-order derivatives with simple multiplication, SINNs can reduce both memory and training time for high-order derivatives.

## 3. Spectral Information Neural Networks (SINNs)

To implement the spectral method on PINNs, the Fourier operator $\mathcal{F}$ is applied to (1), converting the solution from the physics domain to the frequency domain. Practically, we use $\mathcal{F}_N$ to represent the $N$th truncated Fourier operator:

$$
\mathcal{F}_N[u](t, x) = \sum_{k=-N/2}^{N/2-1} \hat{u}(t, k) e^{ikx}, \tag{4}
$$

where $\hat{u}$ corresponds to the Fourier coefficients of $u$, and $i \equiv \sqrt{-1}$ is the unit imaginary number. One can easily

---

[1]Further details are presented in Appendix A
[2]Further details are presented in Appendix B

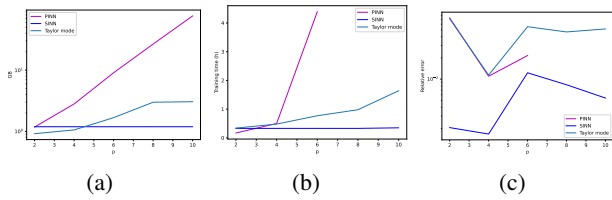

(a)       (b)       (c)

*Figure 1.* In the experiment for (19) with different $p$. (a) depicts the measured maximum memory of the GPU allocated during training, implying that the memory increases exponentially with $p$ for the spatial derivative $\partial^p u / \partial x^p$ in PINNs and is constant for the multiplication $k^p \hat{u}$ in SINNs. Figures (b) and (c) present the training time and relative error of SINNs in comparison with PINNs with and without the Taylor mode. These results indicate that when handling high-order derivatives, SINNs are more efficient and accurate. The further discussion of the experiments is presented in Section 4.2.

obtain the following derivatives:

$$\partial_t \mathcal{F}_N[u](t,x) = \sum_{k=-N/2}^{N/2-1} \partial_t \hat{u}(t,k) e^{ikx}, \qquad (5)$$

$$\partial_x \mathcal{F}_N[u](t,x) = \sum_{k=-N/2}^{N/2-1} ik\hat{u}(t,k) e^{ikx}. \qquad (6)$$

(6) straightforwardly reveals the vanishment of the AD for spacial derivatives in the spectral domain.

Subsequently, we study a two-dimensional (2-D) heat equation a representative example to demonstrate how SINNs work intuitively. Note that, for non-linear equations, the non-linear term produces aliasing error (Gottlieb & Orszag, 1977) which is practically solved by pseudo-spectral methods. We also demonstrate how SINN solves non-linear terms in Appendix C by Navier-Stokes equations. Because SINNs use the Fourier operator, the PDEs to be solved have to be periodic boundary conditions. We further discuss this choice in Section 3.4

### 3.1. SINN for heat equations

Ideally, the heat transfer can be described by the heat equation which is investigated widely in mathematics as one of the prototypical PDEs. Given $\Omega \subset \mathbb{R}^2$, consider the 2-D heat equation with periodic boundary condition:

$$\partial_t u(t,\boldsymbol{x}) = \partial_{xx} u(t,\boldsymbol{x}) + \partial_{yy} u(t,\boldsymbol{x}), \quad t \in [0,T], \ \boldsymbol{x} \in \Omega,$$
$$u(0,\boldsymbol{x}) = \boldsymbol{g}(\boldsymbol{x}), \quad \boldsymbol{x} \in \Omega. \qquad (7)$$

For (7), the residual loss $\mathcal{L}_r(\theta)$ of (2) is explicitly expressed by:

$$\mathcal{L}_r(\theta) = \frac{1}{N_r} \sum_{i=1}^{N_r} \left| \partial_t u^\theta \left( t_r^i, \boldsymbol{x}_r^i \right) \right.$$
$$\left. - \partial_{xx} u^\theta \left( t_r^i, \boldsymbol{x}_r^i \right) - \partial_{yy} u^\theta \left( t_r^i, \boldsymbol{x}_r^i \right) \right|^2, \qquad (8)$$

In our SINNs, the loss function is in the spectral domain without the boundary constraint due to the periodic feature of the Fourier basis. Thus the loss function $\mathcal{L}(\theta)$ is transferred to:

$$\tilde{\mathcal{L}}(\theta) = \tilde{\mathcal{L}}_{ic}(\theta) + \tilde{\mathcal{L}}_r(\theta), \qquad (9)$$

where

$$\tilde{\mathcal{L}}_r(\theta) = \frac{1}{N_r} \sum_{i=1}^{N_r} \left| \partial_t \hat{u}^\theta \left( t_r^i, \boldsymbol{k}^i \right) \right.$$
$$\left. + \left( k_x^i \right)^2 \hat{u}^\theta \left( t_r^i, \boldsymbol{k}^i \right) + \left( k_y^i \right)^2 \hat{u}^\theta \left( t_r^i, \boldsymbol{k}^i \right) \right|^2,$$

$$\tilde{\mathcal{L}}_{ic}(\theta) = \frac{1}{N_{ic}} \sum_{i=1}^{N_{ic}} \left| \hat{u}^\theta \left( 0, \boldsymbol{k}^i \right) - \boldsymbol{g} \left( \boldsymbol{k}^i \right) \right|^2,$$

$$(10)$$

and $\boldsymbol{k}^i = (k_x^i, k_y^i) \in [-N/2, N/2 - 1]^2$ is the sampled frequency from spectral domain. Here we use $\tilde{\mathcal{L}}$ to refer to the loss function associated with the spectral form.

### 3.2. Importance optimization

Compared to PINNs, the main divergence is the importance of different input points. Although the literature on sampling method (Tang et al., 2023; Wu et al., 2023; Lu et al., 2021) shows that the importance of the input points in the physics domain can be dependent on the corresponding residual. Generally speaking, every point is equally important without any prior knowledge. But for SINNs, normally the importance decreases as the corresponding frequency $\boldsymbol{k}$ increases. For instance, the energy spectrum in the inertial ranges of 2-D turbulence (described by the 2-D NS equation) satisfies the scaling relation (Kraichnan, 1967):

$$\sum_{n-\frac{1}{2} \leq |\boldsymbol{k}| < n+\frac{1}{2}} |\hat{\boldsymbol{u}}(t,\boldsymbol{k})|^2 \sim n^{-3}. \qquad (11)$$

Similar physical analysis can be performed for 1-D problems, and the physical background demonstrates that the Fourier coefficient $\hat{\boldsymbol{u}}$ decreases rapidly as an increase in frequency due to the effect of viscosity. In practical, comprehensive experiments in Fourier Neural Operator (Li et al., 2020) show that the truncated Fourier modes can contain most features of the PDEs. Herein, PINNs learn every input point equally, while SINNs are supposed to learn the low-frequency points preferentially. To train the network based on the aforementioned divergence, we propose the sampling method by prior importance:

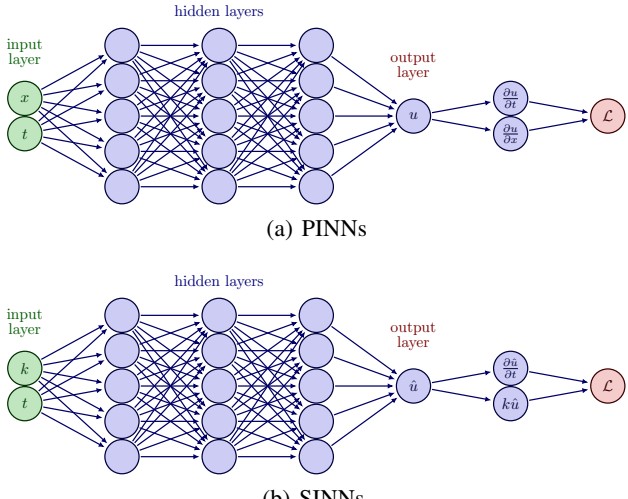

(a) PINNs

(b) SINNs

*Figure 2.* Comparison of PINNs and SINNs. In PINNs, the input is the spatial and temporal grid points $(x, t)$ from the domain, the output is the physical solution $u(x, y)$. In SINNs, the input is the frequency of the Fourier basis $k$ and the temporal grid points $t$, the output is the coefficients $\hat{u}(k, t)$ in the spectral domain. Refer to (4), the derivative $\frac{\partial u}{\partial t}$ is replaced by $\frac{\partial \hat{u}}{\partial t}$, and $\frac{\partial u}{\partial x}$ is replaced by $k\hat{u}$.

Suppose $p(\boldsymbol{k})$ is the probability density function (PDF) used to sample the residual points, we define $p(\boldsymbol{k})$ in SINNs :

$$p(\boldsymbol{k}) \propto \tanh\left[\alpha\left(\|\boldsymbol{k}\|_{\mathrm{mix}}\|\boldsymbol{k}\|_{\infty}^{-\gamma} - N^{1-\gamma}\right)\right], -\infty \leq \gamma < 1, \tag{12}$$

where $\alpha, \gamma, N$ are hyperparameters, $\|\boldsymbol{k}\|_{\mathrm{mix}} = \Pi_{j=1}^{d} \max\{1, n_j\}$. $p(\boldsymbol{k})$ makes SINNs sample more points on the low frequencies and less points on the high frequencies and is shown to be valid in sparse spectral methods for high-dimensional problems, named optimized hyperbolic cross (Shen & Yu, 2010). To demonstrate its distribution intuitively, in Figure 3, we demonstrate how $\gamma$ influence $p(\boldsymbol{k})$ where $\boldsymbol{k} \in [0, 51]^2$.

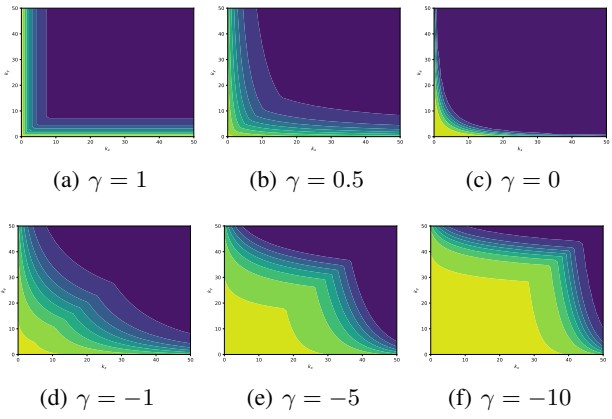

(a) $\gamma = 1$    (b) $\gamma = 0.5$    (c) $\gamma = 0$

(d) $\gamma = -1$    (e) $\gamma = -5$    (f) $\gamma = -10$

*Figure 3.* With the $\gamma$ increases, the values of $p(\boldsymbol{k})$ near the diagonal will become larger. Moreover, the closer a point is to the diagonal line, the more significant the magnitude of this increase becomes. In this demonstration, $\alpha = 10, N = 55$.

### 3.3. Spectral convergence

Regardless of the convergence analysis in temporal domain[3], assume that the capability of MLP is powerful enough, $u \in C^{\infty}(\Omega, \mathbb{R})$ is a smooth function from a subset $\Omega$ of a Euclidean $\mathbb{R}$ space to a Euclidean space $\mathbb{R}$, and $N$ is the number of discretized points.

Firstly, let's review the convergence rate of PINNs. Suppose $u^*$ is the exact solution in the domain $\Omega$ and

$$\theta^* \triangleq \arg\min_{\theta} \int_{\Omega} \mathcal{L}_r\left[u^{\theta}(x)\right] \mathrm{d}x,$$
$$\theta_N^* \triangleq \arg\min_{\theta} \sum_{i=1}^{N} \mathcal{L}_r\left[u^{\theta}(x_i)\right]. \tag{13}$$

Then

$$\|u^{\theta_N^*} - u^*\|_{\Omega} \leq \underbrace{\|u^{\theta_N^*} - u^{\theta^*}\|_{\Omega}}_{\text{statistical error}} + \underbrace{\|u^{\theta^*} - u^*\|_{\Omega}}_{\text{approximation error}}, \tag{14}$$

where approximation error depends on the capability of PINNs. As the capability of MLP is powerful enough, $\|u^{\theta^*} - u^*\|_{\Omega} \ll \|u^{\theta_N^*} - u^{\theta^*}\|_{\Omega}$. Additionally, based on the Monte Carlo method, the statistical error is $\mathcal{O}\left(N^{-1/2}\right)$ (Quarteroni et al., 2006), then:

$$\|u^{\theta_N^*} - u^*\|_{\Omega} = \mathcal{O}\left(N^{-1/2}\right). \tag{15}$$

As for SINNs, with $N$ discretized points, the truncated $u^*$

---

[3]Generally, the temporal error is much smaller than spatial error so we ignore the temporal error in this analysis.

is $u_N^* = \sum_{k=-N/2}^{N/2-1} \hat{u}^*(k)e^{ikx}$, suppose

$$\tilde{\theta}_N^* \triangleq \arg\min_\theta \sum_{i=1}^N \tilde{\mathcal{L}}_r\left[\hat{u}^\theta(k_i)\right]. \tag{16}$$

Then

$$\|u^{\tilde{\theta}_N^*} - u^*\|_\Omega \le \|u^{\tilde{\theta}_N^*} - u_N^*\|_\Omega + \underbrace{\|u_N^* - u^*\|_\Omega}_{\text{spectral error}} \tag{17}$$

As the capability of MLP is powerful enough, $\|u^{\tilde{\theta}_N^*} - u_N^*\|_\Omega \le \sum_{k=-N/2}^{N/2-1} \|\hat{u}^{\tilde{\theta}_N^*}(k) - \hat{u}^*(k)\| = 0$. Furthermore, as the spectral error is exponential convergence (Canuto et al., 1988), then:

$$\|u^{\tilde{\theta}_N^*} - u^*\|_\Omega = o(N^{-s}), \quad \forall s > 0. \tag{18}$$

Thus, the convergence rate of SINNs is $o(N^{-s})$ for any $s > 0$ while the convergence rate of PINNs is $\mathcal{O}(N^{-1/2})$.

### 3.4. The basis function of SINNs

In principle, any basis function can be accommodated in SINNs, especially when dealing with non-periodic problems, Fourier basis functions are not a reasonable choice. However, an exhaustive exploration of all possible basis functions is neither crucial nor necessary in this paper. Therefore, we primarily focus on the Fourier basis function, and the reasons are as follows:

1. Fourier basis function has Faster Fourier Transform (FFT) which can reduce the complexity of computing operator $\mathcal{F}$.

2. Due to the FFT, researchers (Shen et al., 2011; Trefethen, 2000) are exploring transformations to transform other basis functions into Fourier basis functions to take advantage of the FFT for fast computation. Herein, after the corresponding transformation of PDEs, Fourier basis functions can transfer to other basis functions.

3. The multiplier of the multiplication operator $k\hat{u}$ in Fourier basis function is exactly the input $k$. For other basis functions, SINNs should derive at least one extra input dataset with the same size as the original input dataset.

4. The conjugate symmetry property of the Fourier transform of real-valued functions reduces the size of the input dataset from $N^d$ to $N(\frac{N}{2} + 1)$.

## 4. Experiments

To demonstrate the performance of the proposed SINNs, we conducted comprehensive experiments including linear and nonlinear equations from 1D to 3D. The details of the equations and training hyperparameters are available in Appendix D, the metric used in our experiments is relative L2 error (Appendix D.3).

As SINN is a network that mainly changes the type the input and output, and the form of loss function, some strategies can be implemented easily in SINNs. We put those experiments in Section 4.5. Herein, for comparison, we choose two representative methods that cannot be used in SINNs: 1) VSPINN (hoon Song et al., 2024) splits high-order PDEs into 1-order PDE systems. As SINNs compute the high-order derivatives in spectral space, VSPINN is not suitable for SINN. 2) gPINN (Yu et al., 2022) enhances PINN by adding a gradient-based regulation. SINN will change the gradient-based regulation to multiplication by a scaler, and any number multiplied by zero is naturally zero, so gPINN isn't suitable for SINN. Thus, we only include VSPINN and gPINN as baselines. The main results are depicted in Figures 4 and 5. To demonstrate the results intuitively, the detailed data including the statistical variances are shown in Tables 7 and 8. Additionally, as SINNs are more efficient than PINNs, we also discussed the training time in Appendix G. Tables 9 and 10 reveals that, under the same hyperparameters of training, SINN can reduce the training time by a maximum of 39.26%.

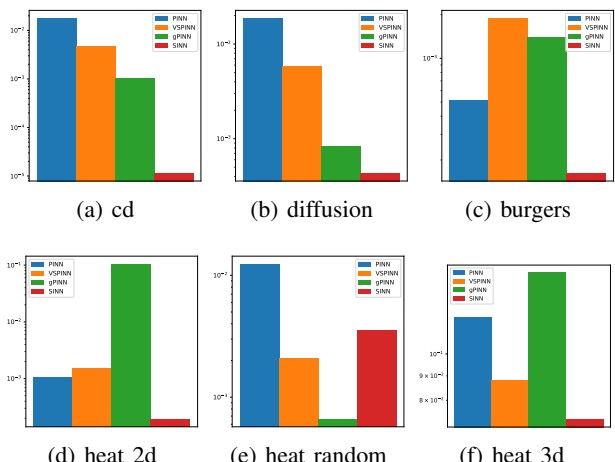

*Figure 4.* We tested divergence experiments including convention-diffusion equations in (a); diffusion equations in (b); heat equations in (d),(e), (f); and burgers' equations in (c).

### 4.1. Preserve physical laws by projections

SINNs can preserve physical laws by simple projections in the spectral domain instead of the soft constraint in the loss function of PINNs. For example, the continuity equation in the Navier-Stokes (NS) equation: $\nabla \boldsymbol{u} = \boldsymbol{0}$ can be preserved by the projection in the spectral domain, in our experiments,

we use the projection $\hat{\boldsymbol{u}} = \left(1 - \frac{\boldsymbol{k}\boldsymbol{k}}{|\boldsymbol{k}|^2}\cdot\right)\hat{\boldsymbol{u}}$. And such kind of strict constraint can obtain a more accurate solutions. The results of Navier-stokes equations are shown in Figure 5 and the predicted solutions and target solutions are depicted in Figure 10. Besides, we also provide the snapshots in Figures 11 and 12.

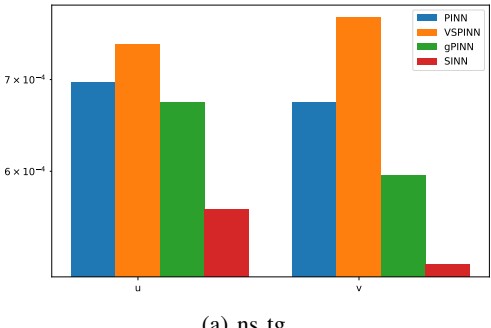

(a) ns_tg

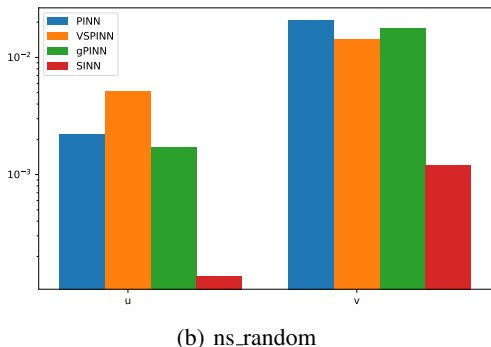

(b) ns_random

*Figure 5.* Navier–Stokes equations (24) describe the motion of viscous fluid substances, and spectral methods have large-scale applications in fluid dynamics (Canuto et al., 1988; Li et al., 2008). our experiments show that SINNs excel PINN and its variants in both classical Taylor–Green vortex Figure 5(a) and randomly generated initial condition (see Appendix D and Figure 10) Figure 5(b).

### 4.2. Different order of derivatives

To demonstrate the efficiency between our SINNs and PINNs, we consider a specific one-dimensional (1-D) hyper-diffusion equation with different order of derivatives:

$$\frac{\partial u}{\partial t} - \epsilon \frac{\partial^p u}{\partial x^p} = 0, \quad x \in [0, 2\pi], t \in [0, T],$$
$$u(0, x) = \sum_{k=0}^{N-1} \sin(kx), \tag{19}$$

where $p$ is the order of the spatial derivatives. To balance the solution of different orders, we set $\epsilon = 0.2^p, T = 0.1$ in our experiments. Taylor mode as a faster method to compute AD by the function *jet* (Bettencourt et al., 2019) is set to be a baseline. The results are shown in Figure 1 and Table 1.

**Training time** One may argue that in Table 1, for the most general derivative term $p = 2$, PINNs are more efficient than SINNs. It is because SINNs have the imaginary part thus the output channels are double the output channels of PINNs. However, we have a more comprehensive comparison in Appendix G, and the conclusion is: if the spacial derivative terms are more than one second-order derivative, including one third-order derivative, or one second-order derivative plus one first-order derivative, our SINNs are more efficient than PINNs; otherwise, PINNs are more efficient.

### 4.3. Different spectral structures

For most problems, the structure of coefficients in the spectral domain is much easier than the structure of solutions in the physical domain. Mathematically speaking, for a dense matrix $U$ discretized from the solution $u(x, y)$, the matrix $\hat{U} = \mathcal{F}U\mathcal{F}^T$ is always sparse. Generally, learning a sparse matrix is easier than learning a dense matrix. Thus to verify that the capability of SINNs is not only for the low-frequency solutions, experiments on the diffusion equation ((20)) are implemented with different $N$:

$$u_t + au_x - \epsilon u_{xx} = 0, \quad x \in [0, 2\pi], t \in [0, T],$$
$$u(0, x) = \sum_{k=0}^{N-1} \sin(kx), \tag{20}$$

where $\epsilon = 0.01, a = 0.1$. Because the analytic solution of (20) is $u(t, x) = \sum_{k=0}^{N-1} \sin(kx - kat)e^{-\epsilon k^2 t}$ which the high-frequency decays exponentially with $t$, we set $T = 0.1$ in this experiment[4]. As we discretize the solution to 100 spatial points, because of the conjugate symmetry property of the Fourier transform of real-valued functions, the maximum $N$ is 51. Thus in our experiments $N = \{5, 7, 9, 15, 23, 30, 50\}$.

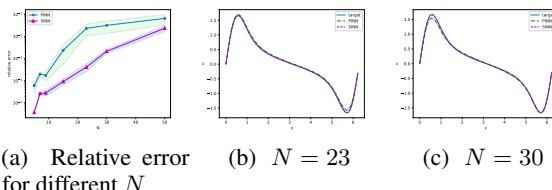

(a) Relative error for different $N$    (b) $N = 23$    (c) $N = 30$

*Figure 6.* In experiments for the diffusion equation (20) with different $N$. The relative error with $N$ is plotted in (a). And the solutions of $N = 23$ and 30 are shown in (b) and (c), respectively. Those results indicate that SINNs are robust even with complex spectral structures. The detailed data are presented in Table 6.

Compared to PINNs, Figure 6 indicates SINNs are more robust even when learning a pretty complex structure in the spectral domain. Notably, although the solution of $N = 30$

---

[4]For long temporal domain experiments, see Tables 3 to 5

*Table 1.* Comparison of the experiments for (19) with different order. '-' means we can't obtain the value because of the limitations of training time.

| | $p$ | Relative error | Training time (hours) | Memory allocated (GB) |
|---|---|---|---|---|
| | 2 | $7.35 \times 10^{-2} \pm 7.07 \times 10^{-4}$ | 0.17 | 1.18 |
| | 4 | $1.10 \times 10^{-2} \pm 7.30 \times 10^{-3}$ | 0.5 | 2.81 |
| PINN | 6 | $2.16 \times 10^{-2} \pm 1.75 \times 10^{-3}$ | 4.39 | 9.06 |
| | 8 | - | - | 26.65 |
| | 10 | - | - | 76.71 |
| | 2 | $7.17 \times 10^{-2} \pm 2.28 \times 10^{-5}$ | 0.34 | 0.92 |
| | 4 | $1.14 \times 10^{-2} \pm 4.32 \times 10^{-5}$ | 0.48 | 1.06 |
| Taylor mode | 6 | $5.53 \times 10^{-2} \pm 7.27 \times 10^{-4}$ | 0.77 | 1.68 |
| | 8 | $4.66 \times 10^{-2} \pm 3.36 \times 10^{-4}$ | 0.98 | 3.00 |
| | 10 | $5.14 \times 10^{-2} \pm 3.73 \times 10^{-4}$ | 1.64 | 3.06 |
| | 2 | $2.06 \times 10^{-3} \pm 1.51 \times 10^{-3}$ | 0.33 | 1.19 |
| | 4 | $1.67 \times 10^{-3} \pm 3.25 \times 10^{-4}$ | 0.33 | 1.19 |
| SINN | 6 | $1.23 \times 10^{-2} \pm 1.36 \times 10^{-3}$ | 0.33 | 1.19 |
| | 8 | $8.33 \times 10^{-3} \pm 1.91 \times 10^{-4}$ | 0.33 | 1.19 |
| | 10 | $5.40 \times 10^{-3} \pm 2.33 \times 10^{-4}$ | 0.35 | 1.19 |

is similar to the solution $N = 23$, PINN also performs worse because of the spectral bias (Xu et al., 2019; Wang et al., 2022b).

### 4.4. Hyperparameters of SI

In this section, we provide an experiment on how to choose the hyperparameters of SI. To tune the hyperparameters, one should consider some prior information from the distribution of the Fourier coefficients: a smaller alpha implies a clearer dividing line and a lower probability of low-importance points being picked; a smaller $N$ means fewer points are regarded as high-importance points; the function of $\gamma$ is shown in Figure 3. We experimented with convection-diffusion equations to show the influence of hyperparameters. The results are shown in Figure 7 and the details including statistical variances can be found in Appendix H. Additionally, We conducted an ablation study on SINNs without SI, and the relative error of heat equation (31) is: $1.51 \times 10^{-3} \pm 1.08 \times 10^{-3}$. Utilizing the best one of SI can increase the accuracy by 89.01%.

### 4.5. Combining SINN with other strategies

Since SINN mainly changes the type of both input and output and the form of the loss function, it can be directly combined with other strategies including the weights for every objective function (Causality (Wang et al., 2022a), NTK (Wang et al., 2021)), the optimizer (L-BFGS (Liu & Nocedal, 1989)) and the architecture (Fourier Embedding(FE) (Tancik et al., 2020), Transformer (Zhao et al., 2023))). Those experiments are conducted on convection-diffusion equations (27) with $\epsilon = 0.5, a = 1, T = 10$.

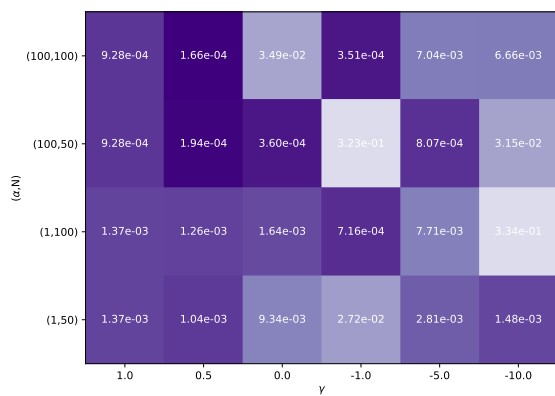

*Figure 7.* Hyperparameters study of SI, the x-axis is $\gamma$, the y-axis is a pair of $(\alpha, N)$. The best hyperparameters are $\alpha = 100, N = 100, \gamma = 0.5$, and the corresponding relative L2 error is $1.66 \times 10^{-4} \pm 1.11 \times 10^{-4}$. This figure also reveals that smaller $\gamma$ and larger $\alpha$ are better for heat equation (31).

The results are shown in Table 2. Although those methods obtain better results, it will take evident extra computation costs.

## 5. Conclusion and future work

In this paper, we propose the Spectral Informed Neural Network (SINN) as a novel approach for solving partial differential equations (PDEs) in the spectral domain. The crucial divergence between the physical domain and the spectral domain is the importance of input points, and therefore we introduce a specialized training strategy tailored

*Table 2.* SINN + strategies

| Method | Relative error |
|---|---|
| SINN | $3.17 \times 10^{-3} \pm 1.45 \times 10^{-3}$ |
| + Causality | $1.72 \times 10^{-3} \pm 7.02 \times 10^{-4}$ |
| + NTK | $2.17 \times 10^{-3} \pm 4.53 \times 10^{-4}$ |
| + FE | $1.46 \times 10^{-3} \pm 5.05 \times 10^{-4}$ |
| + Transformer | $2.82 \times 10^{-3} \pm 3.87 \times 10^{-4}$ |
| + L-BFGS | $2.61 \times 10^{-3} \pm 5.78 \times 10^{-3}$ |

to the features of the spectral domain. The chosen Fourier basis and the faster Fourier transform help us compute the spatial derivatives as well as train the neural network with remarkable efficiency and low memory allocation. Moreover, the inherent exponential accuracy of the spectral method endows SINNs with superior capabilities for solving PDEs. To validate the performance of SINNs, we conducted a comprehensive series of experiments on both linear and nonlinear PDEs. The results of these experiments serve as concrete evidence that SINNs not only substantially reduce training time but also bring about significant improvements in solution accuracy. This makes SINNs a highly promising tool for scientific research and engineering applications where efficient and accurate PDE solving is of paramount importance.

**Limitations** The current SINNs also inherit the disadvantages of spectral methods, for some PDEs with complex geometries or detailed boundaries in more than one space variable would cause spectral methods to collapse, and so would SINNs.

**Future** Apart from the positive results shown in this paper, the above limitations remain to be investigated further in the future. For the inherited disadvantages from spectral methods, in essence, the spectral method is a specific type of collocation methods that rely on selected basis functions satisfying boundary conditions. Similar to the spectral methods, the collocation methods ensure the residual of the target equation approaches zero at the collocation points associated with the basis functions. Therefore, the SINNs could be developed based on the valuable insights of earlier studies (Canuto et al., 1988) on collocation methods.

Furthermore, some tricks in classical spectral methods can be investigated, for example,

1. Compressive Sampling (Candès et al., 2006; Bayındır, 2016) can be studied on SINNs to further reduce the training time.

2. The smoothed series $S_N[u](t, x) =$

$\sum_{k=-N/2}^{N/2-1} \sigma_k \hat{u}(t, k) e^{ikx}$ can help SINNS to handle PDEs with discontinuous solutions or sharp transitions.

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

## A. Faà di Bruno's formula

Given $f, g$ are sufficiently smooth functions, Faà di Bruno's formula states that the $n$-th derivative of $f(g(x))$ is a sum of products of various orders of derivatives of the component functions:

$$\frac{d^p}{dx^p} f(g(x)) = (f \circ g)^{(p)}(x) = \sum_{\pi \in \Pi} f^{(|\pi|)}(g(x)) \cdot \prod_{B \in \pi} g^{(|B|)}(x),$$

where $\Pi$ is the set of all partitions of the set $\{1, \ldots, p\}$, $|\pi|$ is the number of blocks in the partition $\pi$, and $|B|$ is the number of elements in the block $B$. Since Faà di Bruno's formula is a generalization of the chain rule used in first-order, one can directly apply this formula to achieve accurate higher-order AD efficiently by avoiding a lot of redundant computations. However, the total number of partitions of an $n$-element set is the Bell number (Lucas, 1990): $\#\{\Pi\} = B_p = \sum_{k=0}^{p} \binom{p}{k} B_k$ which still increases exponentially with the differentiation order $n$.

## B. Taylor mode

Here we consider computing the derivative of $f \circ g$ (*i.e.* one-layer networks). One may derive the derivative for multi-layer networks by recursion. Consider the polynomial function $x : \mathbb{R} \to \mathbb{R}^n$

$$x(t) = x_0 + x_1 t + \frac{1}{2!} x_2 t^2 + \cdots + \frac{1}{d!} x_d t^d. \tag{21}$$

For a sufficiently smooth vector-valued function $f : \mathbb{R}^n \to \mathbb{R}^m$, suppose $y = (f \circ x)(t) :: \mathbb{R} \to \mathbb{R}^m$, and the truncated Taylor polynomial is

$$y(t) = y_0 + y_1 t + \frac{1}{2!} y_2 t^2 + \cdots + \frac{1}{d!} y_d t^d. \tag{22}$$

obviously, $y_i = \frac{d^i}{dt^i}(f \circ x)(t)$. However, if $x(t)$ is a Taylor polynomial, *i.e.* $x_i = \frac{d^i}{dt^i} x(t)$, then we can derive that:

$$
\begin{aligned}
y_0 &= f(x0) \\
y_1 &= f'(x0)\frac{dx}{dt} = f'(x_0)x_1 \\
y_2 &= f'(x0)\frac{d^2x}{dt^2} + f''(x_0)\left(\frac{dx}{dt}\right)^2 = f'(x_0)x_2 + f''(x_0)x_1^2 \\
&\vdots
\end{aligned}
\tag{23}
$$

The above expansions exactly correspond to the expressions for the higher order derivatives given by Faà di Bruno's formula. Herein, one can use the coefficients of $y(t)$ as derivative coefficients. This method has already been implemented by $jax.experimental.jet$ module.

## C. Loss functions for Navier-Stokes equations

The mathematical model describing incompressible flows of turbulence problems is the incompressible NS equation, namely,

$$\nabla \cdot \boldsymbol{u} = 0, \quad t \in [0, T], \ \boldsymbol{x} \in \Omega, \tag{24a}$$

$$\partial_t \boldsymbol{u} + \boldsymbol{u} \cdot \nabla \boldsymbol{u} = -\nabla p + \nu \triangle \boldsymbol{u}, \quad t \in [0, T], \ \boldsymbol{x} \in \Omega, \tag{24b}$$

$$\boldsymbol{u}(0, \boldsymbol{x}) = \boldsymbol{g}(\boldsymbol{x}), \quad \boldsymbol{x} \in \Omega, \tag{24c}$$

where $\nabla = (\partial_x, \partial_y)$ is the gradient operator, $\boldsymbol{u}(t, \boldsymbol{x}) = (u, v)$ is the hydrodynamic velocity, $p(t, \boldsymbol{x})$ is the mechanical pressure, $\triangle = \nabla \cdot \nabla$ is the Laplace operator, and $\nu$ is kinematic viscosity. By Fourier transform and appropriate derivation tricks (see Appendix F), the continuity equation (24a) and momentum equation (24b) can be expressed in the spectral domain:

$$\boldsymbol{k} \cdot \hat{\boldsymbol{u}} = 0, \tag{25a}$$

$$\partial_t \hat{\boldsymbol{u}} = -\left(1 - \frac{\boldsymbol{k}\boldsymbol{k}\cdot}{|\boldsymbol{k}|^2}\right)\hat{\boldsymbol{N}} - \nu|\boldsymbol{k}|^2\hat{\boldsymbol{u}}, \tag{25b}$$

where $|\boldsymbol{k}|^2 = \boldsymbol{k} \cdot \boldsymbol{k}$ is the inner product for $\boldsymbol{k} = (k_x, k_y)$, and $\hat{\boldsymbol{N}}$ is the non-linear term in the spectral domain, which has the rotational form $\boldsymbol{N} = (\boldsymbol{\nabla} \times \boldsymbol{u}) \times \boldsymbol{u}$ in the physical space (Canuto et al., 1988). The continuity equation (25a) can be preserved strictly by the projection in our SINNs; however, the non-linear term has the challenge of dealing with the aliasing error which is solved by pseudo-spectral methods. To deal with the aliasing error, the loss function is

$$\tilde{\mathcal{L}}_r(\theta) = \frac{1}{N_r} \sum_{i=1}^{N_r} \left| \mathcal{F}^{-1}[\partial_t \hat{\boldsymbol{u}}\left(t_r^i, \boldsymbol{k}\right)] + \mathcal{F}^{-1}\left[\left(1 - \frac{\boldsymbol{k}\boldsymbol{k}\cdot}{|\boldsymbol{k}|^2}\right) \hat{\boldsymbol{N}}\left(t_r^i, \boldsymbol{k}\right)\right] + \nu \mathcal{F}^{-1}\left[|\boldsymbol{k}|^2 \hat{\boldsymbol{u}}\left(t_r^i, \boldsymbol{k}\right)\right] \right|^2. \tag{26}$$

## D. Details of Experiments

In the following experiments, we proceed by training the model via stochastic gradient descent using the Adam (Kingma & Ba, 2014) optimizer with the exponential decay learning rate. The hyperparameters for exponential decay are: the initial learning rate is $10^{-3}$, the decay rate is $0.95$ and the number of transition steps is $10000$. The MLP is equipped with the Sigmoid Linear Unit (SiLU) activations and Xavier initialization.

Note that there is no injection of external source terms in our experiments, resulting in a decay of the quantities, including temperature and energy, over time. As time increases, the related functions gradually become smoother, and the overall flow field tends to be constant. Herein, to distinctly demonstrate the advantages and performance of our SINNs, the temporal domain is restricted to the interval when the flow field undergoes significant changes. Additionally, we experimented in a long temporal domain for the 1-D convection-diffusion equation with periodic boundary conditions.

### D.1. The experiments on linear equations

#### D.1.1. 1-D PROBLEMS

In 1-D problems, we discretize the spatial domain to 100 points and the temporal domain to 100 points, thus the total size of the discretization for PINNs is $100 \times 100$. Thanks to the symmetric of real functions in the spectral domain, the total size of the discretization for SINNs is $51 \times 100$. The MLP we used for both PINNs and SINNs is $10 \times 100$: 10 layers and every hidden layer has 100 neurons. We train both PINNs and SINNs for $5 \times 10^5$ iterations.

**Convection-diffusion equation (cd)**  Our first experiment is the 1-D convection-diffusion equation with periodic boundary conditions, and the convection-diffusion equation can be expressed as follows:

$$u_t + au_x - \epsilon u_{xx} = 0, x \in [0, 2\pi], t \in [0, T],$$
$$u(0, x) = \sum_{k=0}^{N-1} \sin(kx), \tag{27}$$

with the analytic solution

$$u(t, x) = \sum_{k=0}^{N-1} \sin(kx - kat) e^{-\epsilon k^2 t}, \tag{28}$$

where $T = 0.1$, $\epsilon = 0.01$, $a = 0.1$, and $N = 6$ in our experiments.

Additionally, to verify our methods on long temporal domain, we did two experiments one is $T = 1$ (discretize to 100 points) and another is $T = 10$ (discretize to 1000 points). The results are in Table 3 and Table 4 respectively.

*Table 3.* Long temporal domain of $T = 1$

| $t$(s) | 0.10 | 0.30 | 0.50 | 0.70 | 0.90 | 1.00 |
|---|---|---|---|---|---|---|
| relative error | $1.43 \times 10^{-5}$ | $1.55 \times 10^{-5}$ | $1.39 \times 10^{-5}$ | $1.44 \times 10^{-5}$ | $1.47 \times 10^{-5}$ | $1.49 \times 10^{-5}$ |

Furthermore, for a more complex problem with larger coefficients $\epsilon = 0.5, a = 1$, we provide the results in Table 5.

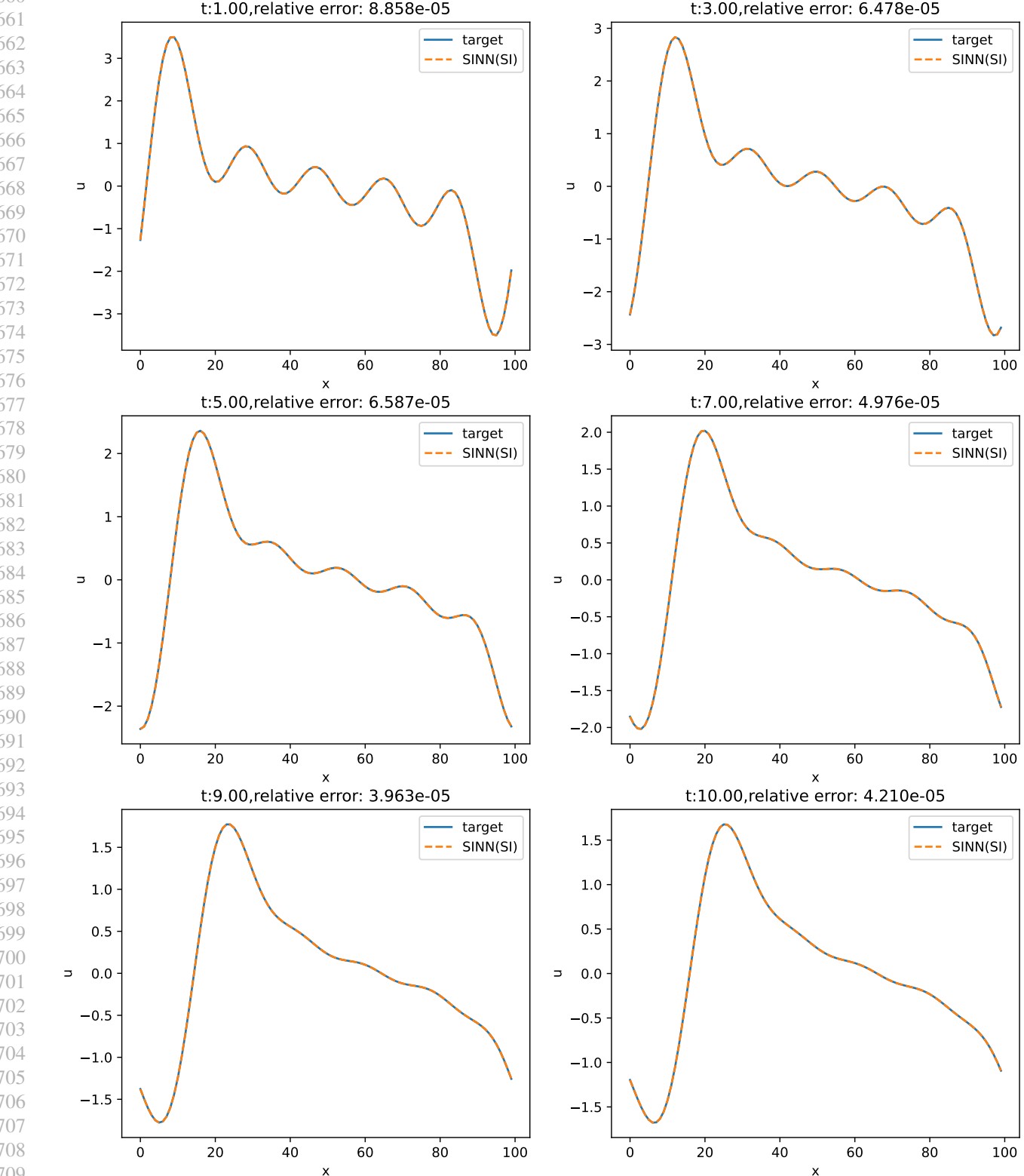

*Figure 8.* Lone temporal domain experiment of $T = 10$, with the time $t$ increases, the function becomes smoother and the relative error becomes even smaller.

*Table 4.* Long temporal domain of $T = 10$

| $t$(s) | 1.00 | 3.00 | 5.00 | 7.00 | 9.00 | 10.00 |
|---|---|---|---|---|---|---|
| relative error | $8.86 \times 10^{-5}$ | $6.48 \times 10^{-5}$ | $6.59 \times 10^{-5}$ | $4.98 \times 10^{-5}$ | $3.96 \times 10^{-5}$ | $4.21 \times 10^{-5}$ |

*Table 5.* A more complex equation on the long temporal domain of $T = 10$

| $t$(s) | 1.00 | 3.00 | 5.00 | 7.00 | 9.00 | 10.00 |
|---|---|---|---|---|---|---|
| relative error | $1.76 \times 10^{-4}$ | $4.80 \times 10^{-4}$ | $1.51 \times 10^{-3}$ | $8.18 \times 10^{-4}$ | $3.39 \times 10^{-3}$ | $6.33 \times 10^{-4}$ |

**Diffusion equation (diffusion)** Another set of experiments on 1-D linear equations is about the diffusion equation, which can be written as

$$u_t = \epsilon u_{xx}, \quad x \in [0, 2\pi], \ t \in [0, T],$$
$$u(0, x) = \sum_{k=0}^{N-1} \sin(kx), \tag{29}$$

with the analytic solution

$$u(t, x) = \sum_{k=0}^{N-1} \sin(kx) e^{-\epsilon k^2 t}, \tag{30}$$

where $T = 0.1$ and $\epsilon = 1.0$ in our experiments. Based on PINNs and SINNs with sampling by importance, Figure 6 illustrates two groups of experiments with varied $N$ Besides, Table 7 shows the experimental results with $N = 23$, while Table 6 presents more results with different $N$.

*Table 6.* Comparison of the relative errors for the diffusion equation with different N in (30)

| $N$ | PINN | SINN |
|---|---|---|
| 5 | $6.07 \times 10^{-5} \pm 3.17 \times 10^{-5}$ | $3.71 \times 10^{-6} \pm 1.59 \times 10^{-7}$ |
| 7 | $2.00 \times 10^{-4} \pm 4.04 \times 10^{-5}$ | $2.66 \times 10^{-5} \pm 1.54 \times 10^{-6}$ |
| 9 | $1.75 \times 10^{-4} \pm 7.42 \times 10^{-5}$ | $2.82 \times 10^{-5} \pm 8.11 \times 10^{-6}$ |
| 15 | $2.38 \times 10^{-3} \pm 1.80 \times 10^{-3}$ | $9.17 \times 10^{-5} \pm 2.62 \times 10^{-5}$ |
| 23 | $2.33 \times 10^{-2} \pm 1.21 \times 10^{-2}$ | $4.15 \times 10^{-4} \pm 1.58 \times 10^{-4}$ |
| 30 | $3.19 \times 10^{-2} \pm 1.80 \times 10^{-2}$ | $2.13 \times 10^{-3} \pm 3.91 \times 10^{-4}$ |
| 50 | $6.55 \times 10^{-2} \pm 3.22 \times 10^{-2}$ | $2.37 \times 10^{-2} \pm 6.89 \times 10^{-3}$ |

One may observe that the weighted loss method fails when $N > 30$. Because the weighted loss forces SINNs to pay more attention to the low-frequency part, SINNs with weighted loss will abandon the high-frequency if the loss on low-frequency is not small enough.

D.1.2. 2-D PROBLEMS

In 2-D problems, the MLP we used for both PINNs and SINNs is $10 \times 100$: 10 layers and every hidden layer has 100 neurons. We train both PINNs and SINNs for $10^6$ iterations.

**Heat equation with analytic solution (heat_2d)** For a 2-D linear problem, the heat equation with the following initial condition is considered here:

$$u_t = \epsilon (u_{xx} + u_{yy}), \quad \boldsymbol{x} \in [0, 2\pi]^2, \ t \in [0, T],$$
$$u(0, \boldsymbol{x}) = \sum_{k=0}^{N-1} [\sin(kx) + \sin(ky)], \tag{31}$$

with the analytic solution

$$u(t, x) = \sum_{k=0}^{N-1} \left[\sin\left(kx\right) + \sin\left(ky\right)\right] e^{-\epsilon k^2 t}, \tag{32}$$

where $T = 0.01$, $\epsilon = 1.0$, and $N = 10$ in our experiment. The discretization of the spatial and temporal domains is set to $100 \times 100$ and 10 points, respectively. Thus, the total size of the discretization for PINNs is $100 \times 100 \times 10$, while the total size for SINNs can be reduced to $51 \times 100 \times 10$ due to the Fourier transform for real functions.

**Heat equation with random initialization (heat_random)** The 2-D heat equation with the Gaussian random initial condition is included here, which can be written as

$$u_t = \epsilon\left(u_{xx} + u_{yy}\right), \quad \boldsymbol{x} \in [0, 2\pi]^2, \ t \in [0, T],$$
$$\hat{u}(0, \boldsymbol{k}) = \hat{g}(\boldsymbol{k}),$$

$$\hat{g}(\boldsymbol{k}) = \begin{cases} 10^4 \sqrt{0.123456/H(1)} h(\boldsymbol{k}), & \dfrac{1}{2} \leq |\boldsymbol{k}| < \dfrac{3}{2}, \\[2mm] 10^4 \sqrt{0.654321/H(2)} h(\boldsymbol{k}), & \dfrac{3}{2} \leq |\boldsymbol{k}| < \dfrac{5}{2}, \\[2mm] 10^4 \sqrt{0.345612/H(3)} h(\boldsymbol{k}), & \dfrac{5}{2} \leq |\boldsymbol{k}| < \dfrac{7}{2}, \\[2mm] 10^4 \sqrt{0.216543/H(4)} h(\boldsymbol{k}), & \dfrac{7}{2} \leq |\boldsymbol{k}| < \dfrac{9}{2}, \\[2mm] 10^4 \sqrt{0.561234/H(5)} h(\boldsymbol{k}), & \dfrac{9}{2} \leq |\boldsymbol{k}| < \dfrac{11}{2}, \\[2mm] 10^4 \sqrt{0.432165/H(6)} h(\boldsymbol{k}), & \dfrac{11}{2} \leq |\boldsymbol{k}| < \dfrac{13}{2}, \\[2mm] 0, & |\boldsymbol{k}| \geq \dfrac{13}{2}, \end{cases} \tag{33}$$

$$H(n) = \sum_{n - \frac{1}{2} \leq |\boldsymbol{k}| < n + \frac{1}{2}} |h(\boldsymbol{k})|^2,$$

where $h \in \mathbb{C}$ generated by standard normal distribution fulfills the symmetry $h(\boldsymbol{k}) = \bar{h}(-\boldsymbol{k})$, and the parameters $T = 0.01$, $\epsilon = 1.0$, and $N = 10$ are taken in our experiment. The spatial and temporal domains are discretized to $100 \times 100$ and 6 points, respectively. The total size of the discretization for PINNs is $100 \times 100 \times 6$, while the total size for SINNs is reduced to $51 \times 100 \times 6$ since the functions are real. The solutions $u(0.01, \boldsymbol{x})$ in our experiments for the 2-D heat equation with the Gaussian random initial condition are plotted in Figure 9. The spectral method in Appendix I computes the numerical solution $u(0.01, \boldsymbol{x})$ with a sufficiently small time step.

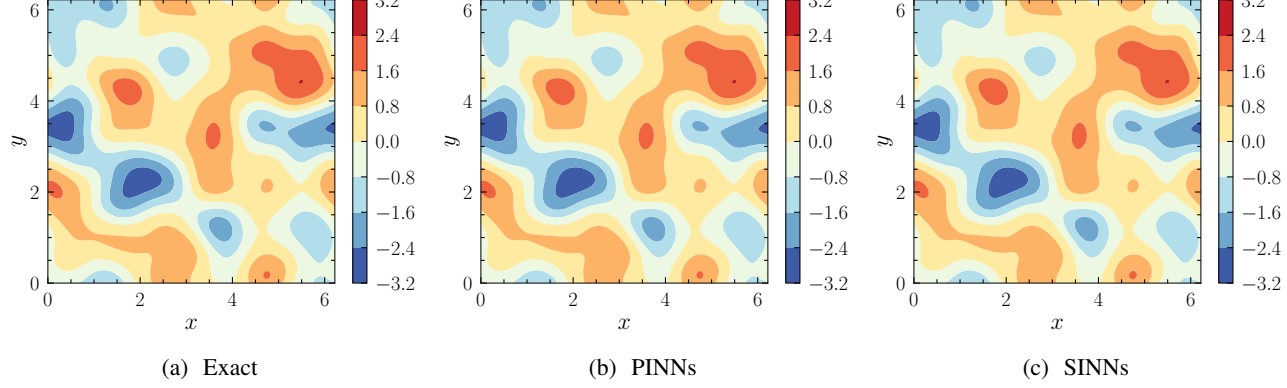

| (a) Exact | (b) PINNs | (c) SINNs |

*Figure 9.* The exact solution, the predicted solution by PINNs, and the predicted solution by SINNs for $u(0.01, \boldsymbol{x})$ in the 2-D heat equation with the Gaussian random initial condition are displayed in (a), (b), and (c), respectively.

### D.1.3. 3-D PROBLEMS

**Heat equation with analytic solution (heat_3d)**   To compare SINNs and PINNs for 3-D linear equations, a heat equation problem is considered here, which has the form

$$u_t = \epsilon \left( u_{xx} + u_{yy} + u_{zz} \right), \quad \boldsymbol{x} \in [0, 2\pi]^3, \ t \in [0, T],$$
$$u(0, \boldsymbol{x}) = \sum_{k=0}^{N-1} \left[ \sin\left(kx\right) + \sin\left(ky\right) + sin\left(kz\right) \right], \tag{34}$$

with the analytic solution

$$u(t, \boldsymbol{x}) = \sum_{k=0}^{N-1} \left[ \sin\left(kx\right) + \sin\left(ky\right) + \sin\left(kz\right) \right] e^{-\epsilon k^3 t}. \tag{35}$$

where $T = 0.01$, $\epsilon = 1.0$, and $N = 5$ in our experiment. The discretized spatial and temporal domains are $100 \times 100 \times 100$ and 10 points, respectively. The total size of the discretization for PINNs is $100 \times 100 \times 100 \times 10$, while the total size for SINNs is decreased to $51 \times 100 \times 100 \times 10$ for the real function $u$.

### D.2. The experiments on non-linear equations

### D.2.1. 1-D PROBLEMS

**Burgers equation (Burgers)**   One of the most important 1-D nonlinear equations is the Burgers equation, taking the following form:

$$u_t = \nu u_{xx} - u u_x, \quad x \in [0, 2\pi], t \in [0, T],$$
$$u(0, x) = \sum_{k=1}^{N} \sin(kx) \tag{36}$$

where $T = 0.1$, $N = 3$ and $\nu = \pi/150$ in our experiment. The discretization of the spatial and temporal domains is set to 100 and 11 points, respectively. The total size of the discretization for PINNs is $100 \times 11$, while the total size for SINNs is $51 \times 11$ for a real $u$.

### D.2.2. 2-D PROBLEMS

**NS equations with Taylor–Green vortex (NS_TG)**   The 2-D nonlinear NS equation is the same as (24) with $T = 2$, $\nu = 2\pi/100$, and $\boldsymbol{g}(\boldsymbol{x}) = \left( -\cos(x)\sin(y), \ \sin(x)\cos(y) \right)$ in our experiment. The spatial and temporal domains are discretized to $100 \times 100$ and 11 points, respectively. The total size for PINNs is $100 \times 100 \times 11$, while the total size for SINNs is $51 \times 100 \times 11$ since $\boldsymbol{u}$ is real.

**NS equations with random initialization (NS_random)**   The 2-D NS equation (24) is also included here with $T = 2$, $\nu = 2\pi/100$, and a random initial condition $\boldsymbol{g}(\boldsymbol{x})$, namely,

$$\hat{\boldsymbol{u}}(0, \boldsymbol{k}) = \hat{\boldsymbol{g}}(\boldsymbol{k}),$$

$$\hat{\boldsymbol{g}}(\boldsymbol{k}) = \begin{cases} 10^4 \sqrt{0.123456/H(1)}(\boldsymbol{h} - \boldsymbol{kk} \cdot \boldsymbol{h}/|\boldsymbol{k}|^2), & \dfrac{1}{2} \leq |\boldsymbol{k}| < \dfrac{3}{2}, \\[2mm] 10^4 \sqrt{0.654321/H(2)}(\boldsymbol{h} - \boldsymbol{kk} \cdot \boldsymbol{h}/|\boldsymbol{k}|^2), & \dfrac{3}{2} \leq |\boldsymbol{k}| < \dfrac{5}{2}, \\[2mm] 10^4 \sqrt{0.345612/H(3)}(\boldsymbol{h} - \boldsymbol{kk} \cdot \boldsymbol{h}/|\boldsymbol{k}|^2), & \dfrac{5}{2} \leq |\boldsymbol{k}| < \dfrac{7}{2}, \\[2mm] \boldsymbol{0}, & |\boldsymbol{k}| \geq \dfrac{7}{2}, \end{cases} \tag{37}$$

$$H(n) = \sum_{n-\frac{1}{2} \leq |\boldsymbol{k}| < n+\frac{1}{2}} |\boldsymbol{h}(\boldsymbol{k})|^2,$$

where $h \in \mathbb{C}$ generated by standard normal distribution fulfills the symmetry $h(\boldsymbol{k}) = \bar{h}(-\boldsymbol{k})$. The spatial and temporal domains are discretized to $100 \times 100$ and 11 points, respectively. And the total size for PINNs is $100 \times 100 \times 11$, while

the total size for SINNs is reduced to $51 \times 100 \times 11$ for a real $\boldsymbol{u}$. The predicted solutions from the SINNs for this problem, including the $x$-component $u$ and $y$-component $v$ of the velocity $\boldsymbol{u}(2, \boldsymbol{x}) = (u, v)$, are plotted in Figure 10. And the corresponding numerical solution $\boldsymbol{u}(2, \boldsymbol{x})$ is obtained with a sufficiently small time step by the spectral method in Appendix I.

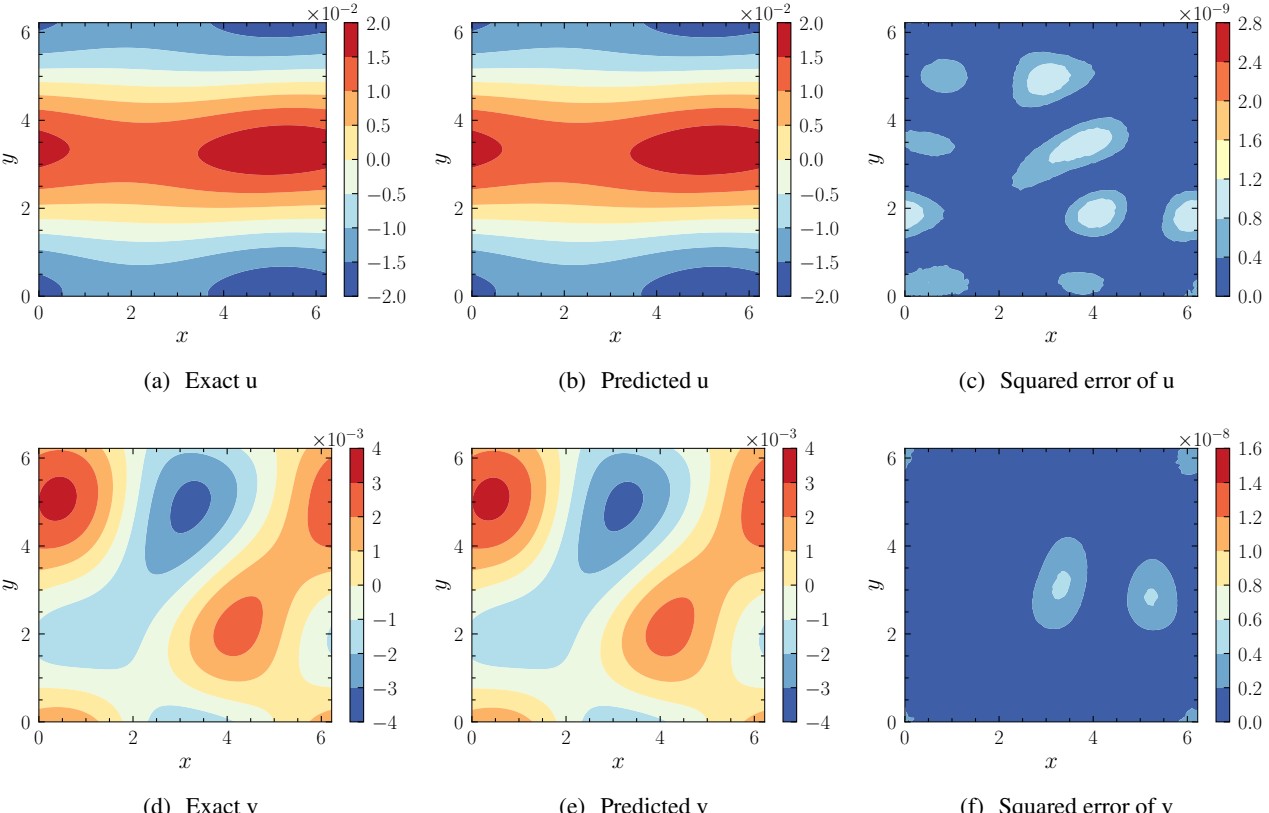

| (a) Exact u | (b) Predicted u | (c) Squared error of u |

| (d) Exact v | (e) Predicted v | (f) Squared error of v |

*Figure 10.* For the 2-D NS equations (24) with the Gaussian random initial condition, exact solutions of $u(2, \boldsymbol{x})$ and $v(2, \boldsymbol{x})$ are plotted on Figures 10(a) and 10(d) respectively, predicted solutions by SINNs of $u(2, \boldsymbol{x})$ and $v(2, \boldsymbol{x})$ are plotted on Figures 10(b) and 10(e) respectively,and the corresponding squared errors are plotted Figures 10(c) and 10(f)

### D.3. The metrics of the relative error

The metric we use is the relative L2 error as follows:

$$E = \frac{\sqrt{\sum_{i=1}^{N} \left| u^{\theta} \left( t^i, x^i \right) - u^T \left( t^i, x^i \right) \right|^2}}{\sqrt{\sum_{i=1}^{N} \left| u^T \left( t^i, x^i \right) \right|^2}}, \tag{38}$$

where $u^T$ is the target solution and $u^{\theta}$ is the trained approximation. In cases where $u^T$ cannot be analytically represented, the spectral method in Appendix I is utilized to obtain high-accuracy numerical solutions.

## E. Results of experiments

## F. Details of the derivation of the spectral form of the incompressible NS equations

This appendix presents the details of the derivation of (25) and the calculation of non-linear terms $\hat{N}$ in spectral space.

Considering the periodic boundary conditions, by applying the Fourier transform on both sides of (24), the NS equations in

*Table 7.* Comparison of the relative errors for the linear equations

| | equation | | PINN | VSPINN | gPINN | SINN |
|---|---|---|---|---|---|---|
| 1-D | cd | $u$ | $1.79 \times 10^{-2} \pm 4.19 \times 10^{-3}$ | $4.77 \times 10^{-3} \pm 1.80 \times 10^{-3}$ | $1.04 \times 10^{-3} \pm 3.80 \times 10^{-4}$ | $1.14 \times 10^{-5} \pm 1.28 \times 10^{-5}$ |
| | diffusion | $u$ | $1.88 \times 10^{-2} \pm 1.44 \times 10^{-2}$ | $5.86 \times 10^{-3} \pm 3.42 \times 10^{-3}$ | $8.40 \times 10^{-4} \pm 1.14 \times 10^{-4}$ | $4.32 \times 10^{-4} \pm 1.61 \times 10^{-4}$ |
| 2-D | heat_2d | $u$ | $1.06 \times 10^{-3} \pm 1.55 \times 10^{-4}$ | $1.51 \times 10^{-3} \pm 1.54 \times 10^{-4}$ | $1.05 \times 10^{-1} \pm 2.18 \times 10^{-3}$ | $1.66 \times 10^{-4} \pm 1.10 \times 10^{-4}$ |
| | heat_random | $u$ | $1.24 \times 10^{-2} \pm 5.24 \times 10^{-3}$ | $2.10 \times 10^{-3} \pm 5.06 \times 10^{-4}$ | $6.62 \times 10^{-4} \pm 3.28 \times 10^{-5}$ | $3.54 \times 10^{-3} \pm 6.65 \times 10^{-4}$ |
| 3-D | heat_3d | $u$ | $1.19 \times 10^{-1} \pm 3.61 \times 10^{-3}$ | $8.80 \times 10^{-2} \pm 1.25 \times 10^{-3}$ | $1.48 \times 10^{-1} \pm 4.71 \times 10^{-5}$ | $7.29 \times 10^{-2} \pm 1.98 \times 10^{-4}$ |

*Table 8.* Comparison of the relative errors for the non-linear equations

| | equation | | PINN | VSPINN | gPINN | SINN |
|---|---|---|---|---|---|---|
| 1-D | Burgers | $u$ | $5.17 \times 10^{-4} \pm 1.50 \times 10^{-4}$ | $1.89 \times 10^{-3} \pm 1.68 \times 10^{-4}$ | $1.39 \times 10^{-3} \pm 9.02 \times 10^{-7}$ | $1.62 \times 10^{-4} \pm 4.70 \times 10^{-5}$ |
| 2-D | NS_TG | $u$ | $6.97 \times 10^{-4} \pm 2.61 \times 10^{-5}$ | $7.43 \times 10^{-4} \pm 2.13 \times 10^{-4}$ | $6.74 \times 10^{-4} \pm 1.10 \times 10^{-4}$ | $5.63 \times 10^{-4} \pm 1.99 \times 10^{-4}$ |
| | | $v$ | $6.74 \times 10^{-4} \pm 3.26 \times 10^{-4}$ | $7.77 \times 10^{-4} \pm 3.18 \times 10^{-4}$ | $5.96 \times 10^{-4} \pm 1.22 \times 10^{-4}$ | $5.13 \times 10^{-4} \pm 1.71 \times 10^{-4}$ |
| | NS_random | $u$ | $2.20 \times 10^{-3} \pm 7.78 \times 10^{-4}$ | $5.15 \times 10^{-3} \pm 3.12 \times 10^{-3}$ | $1.72 \times 10^{-3} \pm 3.96 \times 10^{-4}$ | $1.35 \times 10^{-4} \pm 2.89 \times 10^{-5}$ |
| | | $v$ | $2.06 \times 10^{-2} \pm 8.58 \times 10^{-3}$ | $1.43 \times 10^{-2} \pm 4.63 \times 10^{-3}$ | $1.76 \times 10^{-2} \pm 5.45 \times 10^{-3}$ | $1.19 \times 10^{-3} \pm 1.30 \times 10^{-4}$ |

the spectral space are expressed as

$$i\boldsymbol{k} \cdot \hat{\boldsymbol{u}} = 0, \tag{39a}$$

$$\partial_t \hat{\boldsymbol{u}} + \hat{\boldsymbol{N}} = -i\boldsymbol{k}\hat{p} - \nu|\boldsymbol{k}|^2 \hat{\boldsymbol{u}}. \tag{39b}$$

The continuity equation reveals that frequency and velocity are orthogonal in spectral space; by taking the frequency dot product on both sides of the momentum equation (39b), the relationship between the pressure and the non-linear term can be obtained,

$$\boldsymbol{k} \cdot \hat{\boldsymbol{N}} = -i|\boldsymbol{k}|^2 \hat{p}. \tag{40}$$

Eliminating the pressure in the momentum equation (39b), the form of (25b) can be finally obtained.

According to the identities in field theory, the non-linear terms in (24b) can be expressed in the form

$$\boldsymbol{N} = \boldsymbol{u} \cdot \nabla \boldsymbol{u} = \nabla(\boldsymbol{u} \cdot \boldsymbol{u}/2) = (\nabla \times \boldsymbol{u}) \times \boldsymbol{u}, \tag{41}$$

where the term $\nabla(\boldsymbol{u} \cdot \boldsymbol{u}/2)$ has no contribution to (25b), and the non-linear term can be simplified as the rotational form (Canuto et al., 1988) $\boldsymbol{N} = (\nabla \times \boldsymbol{u}) \times \boldsymbol{u}$. The specific calculation of non-linear terms in spectral space can be written as

$$\boldsymbol{N} = \mathcal{F}^{-1}[i\boldsymbol{k} \times \hat{\boldsymbol{u}}(t, \boldsymbol{k})] \times \mathcal{F}^{-1}[\hat{\boldsymbol{u}}(t, \boldsymbol{k})], \tag{42}$$

## G. Comparison of training time

To briefly demonstrate the comparison of training time, Tables 9 and 10 only shows the training time of PINN and SINN. Because the training time of both VSPINN and gPINN is higher than PINN for all our experiments.

*Table 9.* Comparison of the training time of the linear equations

| | equation | PINN | SINN |
|---|---|---|---|
| 1-D | convection-diffusion[†] | 1.92 h | 1.63 h |
| | diffusion[†] | See Table 6 | |
| 2-D | heat[†] | 8.67 h | 5.28 h |
| | heat_random[†] | 7.87 h | 4.78 h |
| 3-D | heat[‡] | 20.94 h | 14.11 h |

*Table 10.* Comparison of the training time of the non-linear equations

|  | equation | PINN | SINN |
|---|---|---|---|
| 1-D | Burgers[†] | 2.03 h | 1.51 h |
| 2-D | NS_TG[‡] | 3.38 h | 2.51 h |
| | NS_random[‡] | 3.40 h | 2.53 h |

Those superscripts are:

†: train on single Tesla V100-SXM2-16GB with CUDA version: 12.3.

‡: train on single NVIDIA A100-SXM4-80GB with CUDA version: 11.2.

## H. Hyperparameters of SI

We study those hyperparameters on 2D heat equations (31) with the same setting in Table 7 on the hyperparameter domain $\alpha = \{1, 100\}, N = \{50, 100\}, \gamma = \{1.0, 0.5, 0.0, -1.0, -5.0, -10.0\}$. The best hyperparameters are $\alpha = 100, N = 100, \gamma = 0.5$, and the corresponding error is $1.66 \times 10^{-4} \pm 1.11 \times 10^{-4}$ (see Table 11).

*Table 11.* Ablation results

| $\gamma/(\alpha, N)$ | (100,100) | (100,50) | (1,100) | (1,50) |
|---|---|---|---|---|
| 1.0 | $9.28 \times 10^{-4} \pm 1.00 \times 10^{-3}$ | $9.28 \times 10^{-4} \pm 1.00 \times 10^{-3}$ | $1.37 \times 10^{-3} \pm 4.27 \times 10^{-4}$ | $1.37 \times 10^{-3} \pm 4.27 \times 10^{-4}$ |
| 0.5 | $1.66 \times 10^{-4} \pm 1.11 \times 10^{-4}$ | $1.94 \times 10^{-4} \pm 1.10 \times 10^{-4}$ | $1.26 \times 10^{-3} \pm 4.64 \times 10^{-4}$ | $1.04 \times 10^{-3} \pm 3.69 \times 10^{-4}$ |
| 0.0 | $3.49 \times 10^{-2} \pm 4.92 \times 10^{-2}$ | $3.60 \times 10^{-4} \pm 3.21 \times 10^{-4}$ | $1.64 \times 10^{-3} \pm 1.12 \times 10^{-3}$ | $9.34 \times 10^{-3} \pm 1.11 \times 10^{-2}$ |
| -1.0 | $3.51 \times 10^{-4} \pm 2.73 \times 10^{-4}$ | $3.23 \times 10^{-1} \pm 4.55 \times 10^{-1}$ | $7.16 \times 10^{-4} \pm 3.24 \times 10^{-4}$ | $2.72 \times 10^{-2} \pm 3.69 \times 10^{-2}$ |
| -5.0 | $7.04 \times 10^{-3} \pm 7.25 \times 10^{-3}$ | $8.07 \times 10^{-4} \pm 6.48 \times 10^{-4}$ | $7.71 \times 10^{-3} \pm 3.44 \times 10^{-3}$ | $2.81 \times 10^{-3} \pm 4.28 \times 10^{-4}$ |
| -10.0 | $6.66 \times 10^{-3} \pm 7.50 \times 10^{-3}$ | $3.15 \times 10^{-2} \pm 4.40 \times 10^{-2}$ | $3.34 \times 10^{-1} \pm 4.71 \times 10^{-1}$ | $1.48 \times 10^{-3} \pm 8.82 \times 10^{-4}$ |

## I. Details of spectral method

Since the analytical solutions to the 1-D Burgers equation (36) as well as the 2-D heat equation (33) and NS equation (37) with random initialization are difficult to obtain, we developed in-house spectral method codes to provide the corresponding numerical solutions instead.

Specifically, the 1-D Burgers equation (36) in frequency space is

$$\hat{u}_t = -\nu k^2 \hat{u} - \mathcal{F}[u u_x], \quad k \in [-N/2, N/2 - 1], t \in [0, T]. \tag{43}$$

And the time derivative $\hat{u}_t$ can be approximated by the optimal third-order total variation diminishing Runge-Kutta scheme (Gottlieb & Shu, 1998), which has the following explicit discrete form:

$$\hat{u}_1 = \hat{u}(t, k) - \Delta t \left\{ \nu k^2 \hat{u}(t, k) + \mathcal{F} \left[ \mathcal{D} \left[ \mathcal{F}^{-1}[\hat{u}(t, k)] \cdot \mathcal{F}^{-1}[ik\hat{u}(t, k)] \right] \right] \right\},$$

$$\hat{u}_2 = \frac{3}{4}\hat{u}(t, k) + \frac{1}{4}\hat{u}_1 - \frac{1}{4}\Delta t \left\{ \nu k^2 \hat{u}_1 + \mathcal{F} \left[ \mathcal{D} \left[ \mathcal{F}^{-1}[\hat{u}_1] \cdot \mathcal{F}^{-1}[ik\hat{u}_1] \right] \right] \right\}, \tag{44}$$

$$\hat{u}(t + \Delta t, k) = \frac{1}{3}\hat{u}(t, \boldsymbol{k}) + \frac{2}{3}\hat{u}_2 - \frac{2}{3}\Delta t \left\{ \nu k^2 \hat{u}(t, k) + \mathcal{F} \left[ \mathcal{D} \left[ \mathcal{F}^{-1}[\hat{u}_2] \cdot \mathcal{F}^{-1}[ik\hat{u}_2] \right] \right] \right\},$$

where $\Delta t$ is the time step, $\mathcal{D}$ is the dealiasing operator based on the Fourier smoothing method (Hou & Li, 2007), and $\hat{u}_1$ and $\hat{u}_2$ are intermediate variables.

Besides, the 2-D heat equation (33) can be written in frequency space as

$$\hat{u}_t = -\epsilon \left( k_x^2 + k_y^2 \right) \hat{u}, \quad \boldsymbol{k} = (k_x, k_y) \in [-N/2, N/2 - 1]^2, \ t \in [0, T]. \tag{45}$$

The optimal third-order total variation diminishing Runge-Kutta scheme (Gottlieb & Shu, 1998) can be employed for the time derivative $\hat{u}_t$, and the explicit discrete forms for $\hat{u}$ are

$$\hat{u}_1 = \hat{u}(t, \boldsymbol{k}) - \epsilon\Delta t \left(k_x^2 + k_y^2\right) \hat{u}(t, \boldsymbol{k}),$$

$$\hat{u}_2 = \frac{3}{4}\hat{u}(t, \boldsymbol{k}) + \frac{1}{4}\hat{u}_1 - \frac{1}{4}\epsilon\Delta t \left(k_x^2 + k_y^2\right)\hat{u}_1, \tag{46}$$

$$\hat{u}(t + \Delta t, \boldsymbol{k}) = \frac{1}{3}\hat{u}(t, \boldsymbol{k}) + \frac{2}{3}\hat{u}_2 - \frac{2}{3}\epsilon\Delta t \left(k_x^2 + k_y^2\right)\hat{u}_2.$$

As for the 2-D NS equation (37), the second-order Adams-Bashforth scheme (Orszag, 1971) is applied to the time discretization, and the explicit discrete system becomes

$$\hat{\boldsymbol{u}}(t + \Delta t, \boldsymbol{k}) = e^{(-\nu|\boldsymbol{k}|^2\Delta t)}\left(1 - \frac{\boldsymbol{k}\boldsymbol{k}\cdot}{|\boldsymbol{k}|^2}\right)\left[\frac{3\Delta t}{2}\hat{\boldsymbol{N}}(t, \boldsymbol{k}) - \frac{\Delta t}{2}e^{(-\nu|\boldsymbol{k}|^2\Delta t)}\hat{\boldsymbol{N}}(t - \Delta t, \boldsymbol{k}) - \hat{\boldsymbol{u}}(t, \boldsymbol{k})\right],$$

$$\hat{\boldsymbol{N}}(t, \boldsymbol{k}) = \mathcal{F}\left[\mathcal{D}\left[\mathcal{F}^{-1}\left[i\boldsymbol{k} \times \hat{\boldsymbol{u}}\left(t, \boldsymbol{k}\right)\right] \times \mathcal{F}^{-1}\left[\hat{\boldsymbol{u}}\left(t, \boldsymbol{k}\right)\right]\right]\right], \tag{47}$$

$$\hat{\boldsymbol{N}}(t - \Delta t, \boldsymbol{k}) = \mathcal{F}\left[\mathcal{D}\left[\mathcal{F}^{-1}\left[i\boldsymbol{k} \times \hat{\boldsymbol{u}}\left(t - \Delta t, \boldsymbol{k}\right)\right] \times \mathcal{F}^{-1}\left[\hat{\boldsymbol{u}}\left(t - \Delta t, \boldsymbol{k}\right)\right]\right]\right],$$

where $\mathcal{D}$ is the dealiasing operator based on the Fourier smoothing method (Hou & Li, 2007).

In our calculations, the time steps for the aforementioned three discrete forms are chosen to be small enough to minimize the impact of numerical errors on the solutions.

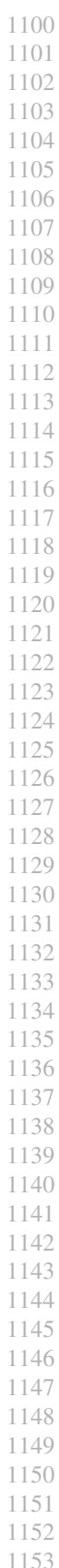

*Figure 11.* Representative snapshots of the predicted $u$ against the ground truth at $t = 0.4, 0.8, 1.2, 1.6$

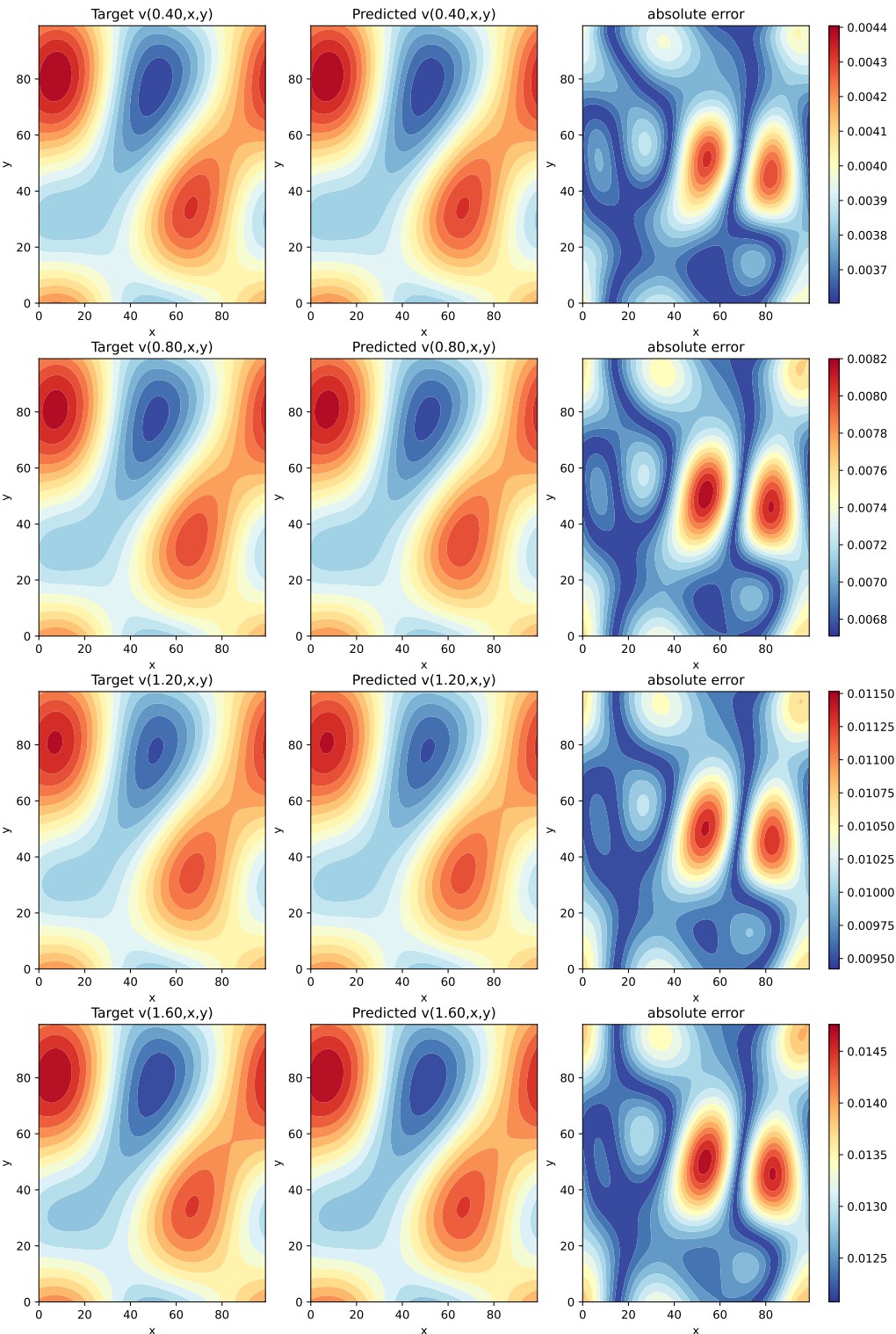

*Figure 12.* Representative snapshots of the predicted $v$ against the ground truth at $t = 0.4, 0.8, 1.2, 1.6$

