# OpenReview forum: "Spectral Informed Neural Networks"
_ICML.cc/2025/Conference — Submitted to ICML 2025_

### Official Review · Reviewer_Swb4 · 2025-03-06

**Overall Recommendation:** 3

**Summary:**

This paper proposes spectral-informed neural networks (SINNs), which solve PDEs using spectral information. SINNs uses less memory than PINNs, especially for problems with higher-order derivatives, and it also obtains better accuracy than PINNs across several experimental settings. The authors also provide a theoretical error convergence analysis to show why SINNs outperform PINNs.

**Claims And Evidence:**

The authors claim that their SINN approach has a lower memory requirement than PINNs, and they clearly show this is true in the paper.

The authors show that SINNs outperform PINNs on several experimental settings, which supports the contributions that they state in the introduction. However I’m concerned that some of the experimental settings in the paper are biased towards the SINN method. For example, the initialization for the 2-D heat and NS equations (see Equations 33 and 37) induce decay in the higher frequency components of the initial condition. If there was no decay, should we still expect SINNs to outperform PINNs?

Moreover, the authors claim to provide an error convergence analysis to show that SINNs are more accurate than PINNs. However, I believe that this claim is not properly justified (see “Theoretical Claims”).

**Essential References Not Discussed:**

The authors should talk about how their method relates to other methods in the literature that combine spectral methods with PINNs. How does their method fit into the prior work combining these two ideas? Here are some papers that have combined these ideas:
1. https://arxiv.org/abs/2202.02710.
2. https://openreview.net/pdf?id=218sl_mPChc.

**Experimental Designs Or Analyses:**

The experimental design appears to be sound. The authors’ clearly describe the details for the experiments in Appendix D.

**Methods And Evaluation Criteria:**

I believe that the proposed methods make sense for the problems being solved in the paper. I also appreciate that the authors thoroughly compare their method to other PINN-based approaches. However, I am not sure if this paper is suitable for the application-driven machine learning track. If I recall correctly, a submission to the application-driven ML track is supposed to address a real-world use case, but the experimental settings in the paper are synthetic.

**Other Comments Or Suggestions:**

The font in several figures is too small to read. I would encourage the authors to increase the font size in these figures.

“scaler” should be “scalar” on page 5

“spacial” should be “spatial”

“burgers” should be “Burgers’”

Top of page 6: there appear to be typos in the expression for $\hat u$

“Navier-stokes” should be “Navier-Stokes” on page 6

The y-axes in Figures 4 and 5 should be labeled. Also, what are u and v in the x-axes in Figure 5?

Equation 23 in Appendix B: $x0$ should be $x_0$

**Other Strengths And Weaknesses:**

A weakness of the work, which is also stated by the authors (to their credit), is that SINNs only apply to PDEs with periodic boundary conditions. This limits the utility of SINNs to a special class of PDEs.

**Questions For Authors:**

1. Page 4: What is $n_j$ in the definition of $\|k\|_{\mathrm{mix}}$?

2. Page 5: I appreciate the authors addressing why they only explore Fourier basis functions in the paper. However, the references that they cite in point 2 are over 10 years old. Is the approach of transforming Fourier basis functions to other basis functions still an active area of research? If not, this could undermine the authors’ justification for only exploring the Fourier basis.

3. Page 6: In Figure 6a, why does the relative error for SINN approach the relative error for PINN as N increases? This seems to contradict the caption for the figure, which says that “SINNs are robust even with complex spectral structures”, making me skeptical of some of the claims in the paper.

**Relation To Broader Scientific Literature:**

The SINN method proposed in the paper is an interesting approach for solving PDEs with periodic boundary conditions. To the best of my knowledge, there is no existing work that has tried out this exact approach for solving PDEs.

**Theoretical Claims:**

In section 3.3, what does it mean to “assume that the capability of MLP is powerful enough”? Is this statement referring to the universal approximation theorem? If so, this statement should be made rigorous.

The $O(N^{-s})$ result for spectral convergence implicitly relies on the fact that we can find the optimum $\tilde{\theta}_N^*$ during optimization. However, this is not a reasonable assumption, since the SINN training problem is non-convex. Perhaps an NTK-style argument could fix some of these issues?

---

> ### Author Rebuttal · Authors · 2025-03-31
>
> Thank you for your extensive feedback. We understand that you have several questions about the paper, and we address each of your points in turn below.
>
> # Questions 1 and Other Comments Or Suggestions
> First and foremost, we apologize for the multiple typos and notational inconsistencies you encountered. We take full responsibility for these and will correct all of them in the final paper. In Figure 5, $u$ and $v$ represent velocity components in $\boldsymbol{u}=(u,v)$ which is the solution of the Navier-Stokes equation. Also, since the notation $k = (n_1, n_2, \dots, n_N)$ was confusing, it is better to use $k= (k_1, k_2, \dots, k_N)$ and $\|\boldsymbol{k}\| _{\text{mix}}=\Pi _{j=1}^d \max  \lbrace 1,k _j\rbrace$.
>
> # Question 2
>
> Transforming Fourier basis functions to other basis functions is a very mature and classical research field. However, only exploring the Fourier basis is not unfair. There are two reasons: 1) it is not practical to explore all basis functions in one paper. 2) exploring one kind of basis is enough. For example, Fourier Neural Operator only explores Fourier basis in Neural Operator, but after FNO, researchers propose Spectral Neural Operator, Wavelet Neural Operator, and so on.
>
> We understand that you are worrying about other basis functions. We conducted a simple example with Chebyshev basis on 1D heat equations:
> $$
> u_{xx}=\epsilon u_t, \quad (x,t)\in[-1,1]\times[0,T],
> $$
> $$
> u(0,x)= \frac{e^{-\epsilon x ^ 2 / 12}} {\sqrt{3}},\quad x \in [-1,1]. \quad u(t,1)=u(t,-1)=\frac{e^{-\epsilon / 12}} {\sqrt{3}}.
> $$
>
> with the solution $u(x,t)=e^{-\frac{\epsilon x ^ 2 } {4 (t + 3)}} / \sqrt{t + 3}$. In our experiments, $T=1.0, \epsilon=10$.
>
> The results are:
> |PINN|SINN|
> |----|----|
> | $3.25\times 10^{-4}\pm1.74\times 10^{-5}$ | $1.17\times 10^{-4}\pm4.61\times 10^{-5}$ |
>
> # Question 3
>
> The $N$ in this experiment is the number of $\sin$ terms in the initial condition of Eq.20. To avoid confusion, it is better to use $N_k$ instead of $N$.
>
> This experiment is conducted to verify the robustness of SINNs. As we stated in Section 4.3, 'For most problems, the structure of coefficients in the spectral domain is simpler than the structure of solutions in the physical domain.', low frequencies can represent most cases in the real world. But how about some specific cases in which the structure of coefficients is also complex?
>
> Furthermore, more experiments on testing high-frequency terms are asked by other reviewers and can answer your concern of 'decay in the higher frequency components of the initial condition'. You can find the experiments in the reviewers of zt1d and 34of.
>
> #  Application-driven ML
>
> When comparing algorithms, precise relative error measurement is essential. Real-world cases are noisy, and full of unknowns, making it hard to isolate and assess algorithm performance. By using synthetic examples, we can better evaluate our algorithm's core capabilities, laying a solid groundwork for real-world applications. For example,
> because of its inherent mathematical complexity, the 2-D Taylor-Green vortex flow has been established as a representative benchmark in computational fluid dynamics for validating numerical methods, fincluding
> finite element penalty–projection method (doi: 10.1016/j.jcp.2006.01.019), PINN (doi: 10.1016/j.jcp.2021.110676), and so on.
>
> Furthermore, the periodic boundary condition is not real, but it can be used to understand the real world's features, for example, coherent vortex evolution (doi: 10.1017/S0022112084001750), intrinsic physical dynamics(doi: 10.1017/S0022112095000012), and so on.
> # Theoretical Claims
> 'Is this statement ..': Yes, the conclusion is held by the universal approximation theorem, thanks for the question.
>
> 'Perhaps an NTK-style ..':  The error analysis is based on a very ideal assumption. NTK analysis is based on the loss function, which is constructed by the residual equation rather than the error equation in both PINNs and SINNs. To our best knowledge, there is no research that reveals that the error and the residual have direct relationships. Furthermore, empirically speaking, a smaller loss function may not guarantee a smaller error (see Error Certification (arXiv:2305.10157)). However,  Ref(doi:10.1016/j.jcp.2023.112527) proves that: adding some terms to the error yields an upper bound including the residual. Thus, if we can find or construct a connection between the residual (or other loss functions, for example, Astral (arXiv:2406.02645)) and the error, perhaps we can use NTK analysis.
>
> # Eq.26
>
> Thanks for being concerned about the aliasing error. How to dealiase the error lies in how to deal with $\hat{N}$.  In a trivial way, one can do inverse FFT back to the physical domain and do those multiplications. Additionally, one can use the 2/3 rule (doi: 10.1175/1520-0469(1971)028<1074:OTEOAI>2.0.CO;2)
>
> # Essential References
>
> Thanks, we will give a further discussion on the relevance of those papers.

---

> > ### Comment · Reviewer_Swb4 · 2025-04-05
> >
> > Thanks for addressing my concerns. I will raise my score accordingly.

---

### Official Review · Reviewer_34of · 2025-03-09

**Overall Recommendation:** 4

**Summary:**

The Manuscript describes how to use  spatial k-space description of the trial neural network with promising results. The general form of the PDE is assumed to be first order in time with periodic boundary conditions in space. Numerous essential PDEs are considered and it is shown that the naturally sparse k-space representations do provide a significant advantage especially for equations with high oder derivatives and allow non-uniform sampling in k-space for the collocation points. The Authors describe a parametric probability distribution for high sampling rate at lower frequencies  diminishing towards the higher frequencies.

**Claims And Evidence:**

The claims correspond to the evidence, also the restrictions of the method are stated clearly, like the difficulty for realistic i.e. (complicated)  domains important in industrial work.

**Essential References Not Discussed:**

The relevant references are cited in the Manuscript.

**Experimental Designs Or Analyses:**

Experiments are well performed,

**Methods And Evaluation Criteria:**

The method does work well for the cases it works well. The challenges  are clearly stated by the Authors, for example the experimental finds the for first order derivates the PINNs are more efficient.

**Other Comments Or Suggestions:**

I would have liked to have more discussion on the Burgers equation with low viscosity, where the hight frequency components become essential. How is this scaling?

**Other Strengths And Weaknesses:**

The Manuscript is well written, and shows very promising results. The basic weakness is, as stated by the Authors,  that as a general PDE solver it is restricted to simple periodic geometries. Hence, its impact is restricted although definitely interesting for the industrial community.

However, a discussion and experimentation of defining initial and boundary conditions from experimental measurements could be considered as a choice. This would allow the estimation, for example,  he material parameters of a PDE.  In this domain and use case,  the experimental measurement can be done in particularly designed geometries that fulfill the periodicity expectations of the SINNs and this would not be such a sin anymore, and would improve the value of the method in practical applications.

**Questions For Authors:**

See above.

**Relation To Broader Scientific Literature:**

The Manuscript as a clear introduction with proper references to the literature positioning the work amongst the physics informed machine dee learning scene.

**Theoretical Claims:**

The theory is sound, in one place when describing continuity equation is k-space the spatial derivative used is gradient when it is supposed to be divergence.

---

> ### Author Rebuttal · Authors · 2025-03-31
>
> Thanks for your positive feedback and useful suggestions.
>
> Firstly, we sincerely appreciate your having caught this typo of $\nabla$ which should be $\nabla$ $\cdot$.
>
> # Burgers’ equation with low viscosity
>
> We already conducted an experiment on Burgers’ equation with a fairly low viscosity ($\mu = \pi/150 \approx 0.0209$) which already contrains high-frequency content. But we agree that exploring different viscosities is insightful.
>
> Here we consider a more explicit Burgers' equation:
>
> $$
> u_t  = \nu u_{xx} - u u_x,  \quad x\in \left[0,2\pi\right], t \in \left[0,T\right]
> $$
>
> $$
> u(0,x) =\sin (x)+\sin(5x)
> $$
>
> which contains a high frequency component and a low frequency component.
>
>  We set $\nu=\pi/r$. The results with different $r$ are as follows:
>
>
> | $r$  | relative error |
> |------|----------------|
> | 20   | $5.23\times10^{-4} \pm 9.48\times10^{-5}$ |
> | 40   | $1.59\times10^{-3} \pm 2.17\times10^{-4}$ |
> | 80   | $9.21\times10^{-3} \pm 6.41\times10^{-3}$ |
> | 160  | $9.23\times10^{-3} \pm 2.98\times10^{-3}$ |
> | 320  | $2.19\times10^{-2} \pm 1.30\times10^{-2}$ |
>
> # Helmholtz
>
> Additionally, Reviewer zt1d suggests conducting experiments on Helmholtz equations which only contain a single frequency.
>
> Here we consider a 1D Helmholtz equation:
>
> $$
>  u_{xx} + \lambda^2u=0, x\in [0,2\pi],
> $$
> with the boundary condition
> $$
> u(0)=u(2\pi)=1, \quad u_x(0)=u_x(2\pi)=\lambda.
> $$
>
> The solution is $u=\cos(\lambda  x)+\sin( \lambda x)$. The results are as follows:
>
> | $\lambda$ | PINNs | SINNs |
> | --- | --- | --- |
> | 2 | $5.37\times10^{-4} \pm 5.30\times10^{-4}$ | $4.72\times10^{-6} \pm 3.39\times10^{-6}$ |
> | 4 | $9.02\times10^{-4} \pm 4.54\times10^{-4}$ | $1.90\times10^{-5} \pm 3.71\times10^{-6}$ |
> | 8 | $1.75\times10^{-3} \pm 6.41\times10^{-4}$ | $1.45\times10^{-4} \pm 1.87\times10^{-4}$ |
> | 16 | $4.80\times10^{-1} \pm 3.38\times10^{-1}$ | $8.63\times10^{-6} \pm 6.24\times10^{-3}$ |
> | 32 | $9.94\times10^{-1} \pm 7.57\times10^{-3}$ | $1.47\times10^{-3} \pm 1.46\times10^{-3}$ |

---

### Official Review · Reviewer_PNMU · 2025-03-13

**Overall Recommendation:** 4

**Summary:**

PINNs have arisen as an exciting and promising alternative to classical solution methods for solving partial differential equations.
However, PINNs are not without their challenges. One key issue is the cost of automatic differentiation for higher-order derivative PDEs. It is well-known that the cost scales with the dimensionality of spatial variables, which is problematic for higher-order PDEs in high dimensions.
To alleviate this issue, the paper proposes Spectral PINNs, which work in spectral space, and reduces applying the differential operator to multiplication, a highly efficient operation on gpu. The authors provide theoretical support to validate their approach, along with experiments to verify its effectiveness in practice.

**Claims And Evidence:**

The paper makes three central claims of contribution:

**Claims:**

1.) We propose a method that eliminates the spatial derivatives of the network to deal with the high GPU memory
consumption of PINNs.

2.) We propose a strategy to approximate the primary
features in the spectral domain by learning the low frequency preferentially to handle the difference between SINNs and PINNs.

3.) We provide an error convergence analysis to show that SINNs are more accurate than PINNs. Furthermore,
our experiments corroborate that the method can reduce the training time and improve the accuracy simultaneously.

**Overall:**

The only claim I really have issue with is item 3.) in particular the first part on error analysis. The assumptions made to reach that conclusions is highly idealized, as I discuss more in detail below.

Given that the claims are mostly well supported and the methods does well in the experiments, I'm rating the submission as a weak accept at this point. My main reason for not rating the submission higher is that spectral methods are somewhat limited in the problem complexity they can handle. In particular, they do not handle complex domain geometries well.

**Essential References Not Discussed:**

I cannot think of any essential references the paper missed in its discussion, so the paper's coverage is sufficient.

**Experimental Designs Or Analyses:**

Yes, I checked and this seems fine. As I said above, it would have been nice to see a comparison with STDE in Table 1 rather than just Taylor mode automatic differentiation. This is my main criticism.

**Methods And Evaluation Criteria:**

The proposed methods and evaluation criteria for the paper are appropriate. As the SINNs framework can be combined with most PINN frameworks/strategies, it makes the most sense to compare to the vanilla PINN and PINN frameworks for which SINNs does not apply.
This is exactly what the paper does.
Moreover, the paper uses standard benchmark PDEs for its evaluation.

Thus, overall I don't have any issues with the baselines for comparison or the experimental design. Though one thing that would have improved the paper is if it had compared with the recently proposed Stochastic Taylor Derivative Estimator (STDE) (Shi et al. 2024) which claims to greatly reduce the cost of differential operators for higher-order PDEs.

**Other Comments Or Suggestions:**

I have no other comments or suggestions at this time.

**Other Strengths And Weaknesses:**

**Strengths**
- The key ideas of the paper are expressed clearly and are easy to follow.

**Questions For Authors:**

Is there any particular reason why you did not compare with STDE?

**Relation To Broader Scientific Literature:**

The paper lies at the intersection of the literature on PINNs and spectral methods for solving PDEs.
Its main proposal is to enhance PINNs with techniques from spectral methods for PDEs to reduce the time required to compute derivatives.
Given the evaluation in the paper, I believe this approach will enhance the use of PINNs for any PDE problem that possesses structure for which it is appropriate to apply a spectral method.
Thus, in terms of the PINNs literature it falls in between papers that propose general strategies to enhance training vs. those who propose problem specific strategies for improving the performance of PINNs.

**Theoretical Claims:**

This is my main issue with the paper. The claims of spectral convergence in section 3.3 is both too informal and oversimplified.
I will focus on the PINN case, but analogous statements hold for SINN.

For instance, in the case of a PINN it decompose the error into the sum of two-terms via the triangle inequality.
1) Statistical Error: $||u^{\theta_N^{\star}}-u^{\theta^{\star}}||_{\Omega}$
2) Approximation Error: $||u^{\theta^{\star}} - u^{\star}||_{\Omega}$.

It then state assuming the neural network is powerful enough, the approximation error term vanishes, while the statistical term is $O(N^{-1/2})$, as the PINN objective is an empirical estimator.
This greatly oversimplifies things.
For one the approximation error is likely small, but treating it as exactly zero is somewhat unreasonable. But my bigger issue, is the entire discussion neglects optimization.
The PINN objective is a challenging non-convex optimization to solve (and similarly for SINNs), so its highly unlikely the solution found via training corresponds to the globally optimal solution of the empirical risk $u^{\theta_N^{\star}}.$
Thus, in general we should expect achieved in practice to be worse than $O(N^{-1/2})$ or $O(N^{-s})$.

I'm sympathetic to the authors in that I understand, analyzing the solution error of PINNs is highly non-trivial, with papers entirely focused on this issue. So, I would be satisfied if the authors were more explicit in their assumptions and that they are considering an ideal setting where optimization error is ignored / negligible.

---

> ### Author Rebuttal · Authors · 2025-03-31
>
> Thank you for your detailed review. We address each of your points below:
> # Is there any particular reason why you did not compare with STDE?
>
> STDE is indeed a recent work to speed up high-order derivatives in PINNs. However, STDE works by amortizing the cost of computing mixed partials in multivariate problems through random estimations. Thus, for univariate problems, its “stochastic” part isn’t needed and it essentially performs the same operations as Taylor-mode. In our manuscript, to intuitively and concisely demonstrate the results, we choose a univariate function with only one spatial derivative term. Herein, Taylor-mode performs the same as STDE. As for multivariate functions, because STDE is an estimator and still uses univariate Taylor-mode, STDE still scales with the order of derivatives and the dimensionality. However, SINNs remain in a constant number with both the order of derivatives and the dimensionality.
>
> To provide a more convinced conclusion, we conducted a test on the efficiency of $d$-variate problems. If  we show SINN is constant with the dimensionality,  combining the result that SINNs is constant with the order of derivative, we can claim that SINNs is more efficient than STDE.
> We consider a Poisson's equation:
> $$
> \Delta u =f, \boldsymbol{x} \in [0,2\pi]^d
> $$
> with $u=\sum_{i=1}^d \cos(x_i)$ and $f=-\sum_{i=1}^d \cos(x_i)$.
> The results are as follows:
> |$d$ |Memory (MB)| Iteration rate (ite/s)|
> |--|------|-------|
> |2|542|992|
> |5|550|1000|
> |10|550|1000|
> |20|790|982|
> |100|8736|995|
>
> The memory increases because the input scales with $d$. But, because our operator is only multiplication, the iteration rate is almost constant as expected.
>
> # Theoretical Claims
> We agree that we are considering an ideal setting that the network can express the truncated Fourier series accurately. And we admit that, in practice, the error we obtained is worse than the theorem results. We will clarify in the paper that the theoretical exponential convergence rate is an asymptotic result assuming sufficient capacity.
>
> Additionally, we don't expect treating the approximation error as exactly zero.  $\| u^{\theta^*}-u^*\| \ll \|u^{\theta^*_N}- u^{\theta^*}\|$ is enough in error analysis.
>
> # Irregular geometries
> Thanks for pointing out the limitation that our experiments were on regular domains.  However, we argue that our SINNs can be extended to irregular geometries with additional operations. There are several approaches to deal with this limitation, for example:
>
> 1) The moving mesh method can construct a coordinate transformation to map an irregular domain to a regular domain (or map subdomains respectively, for example, doi: 10.1016/j.jcp.2020.109835), and it has already been used in PINNs by PhyGeoNet (Gao et al., 2021). (impressive results in Fig. 5 and 7).
>
> 2) One can extend the irregular domain to the regular domain, for example,  the fictitious domain method(doi: 10.1137/20M1345153).
>
> 3) Even, there is a simpler and more straightforward approach if the irregular region can be expected to be expanded into a regular region: one can just add the irregular geometries as a constraint by inverse Fourier transformation. Mathmatically speaking, for PINNs: we minimize the boundary loss by $\mathcal{L} [u_{NN}] (x_b)$, where $u_{NN}$ is the output; For SINNs: we minimize the boundary loss by $\mathcal{L} [\mathcal{F}^{-1} [\hat{u}_{NN}]] (x_b)$, where $\hat{u} _{NN}$ is the output.
>
> Here we consider a 2D Helmholtz equation on a plate with a hole using the third approach:
> $$
> \Delta  u+ \lambda^2u=0, \boldsymbol{x}\in \Omega = [0,2\pi]^2\textbackslash B_\pi(\boldsymbol{x}),
> $$
> with Dirichlet boundary condition:
> $$
> u(\boldsymbol{x})=g(\boldsymbol{x}), \boldsymbol{x} \in \partial \Omega,
> $$
> where $B_\pi(\boldsymbol{x})$ is the ball with the center at the origin (0,0) and a radius of $\pi$ and $g$ is derived by the solution $u=(\cos(\lambda x)+\sin(\lambda x))(\cos(\lambda y)+\sin(\lambda y))$.
>
> Here are the results:
> |$\lambda$| PINN | SINN|
> |---|----------------|----------------|
> |1|$3.59\times10^{-4} \pm 3.11\times10^{-4}$ | $5.31\times10^{-7} \pm 2.11\times10^{-7}$ |
> |2|$2.20\times10^{-2} \pm 6.09\times10^{-3}$ | $2.15\times10^{-5} \pm 2.87\times10^{-5}$ |

---

> > ### Comment · Reviewer_PNMU · 2025-04-05
> >
> > I thank the authors for addressing my concerns, and so will raise my score to a 4.

---

### Official Review · Reviewer_zt1d · 2025-03-14

**Overall Recommendation:** 3

**Summary:**

This paper introduces Spectral-Informed Neural Networks (SINNs) as an alternative to standard Physics-Informed Neural Networks (PINNs) for solving PDEs. Instead of computing spatial derivatives through automatic differentiation, SINNs leverage spectral methods, replacing differentiation with simple multiplications in the frequency domain. The authors claim that this approach significantly reduces memory consumption and computational cost while improving accuracy due to spectral methods' exponential convergence properties. The paper presents experimental results demonstrating efficiency gains over PINNs, particularly for higher-order PDEs.

**Claims And Evidence:**

While the proposed method presents an interesting alternative to standard PINNs, some of its key claims lack sufficient evidence:

- One of the well-known advantages of PINNs is their ability to handle complex geometries flexibly. However, the spectral method used in this paper is inherently tied to structured grids and periodic domains, making it unsuitable for irregular geometries. The paper does not provide any solution or discussion on overcoming this limitation.

- The spectral approach reduces memory consumption for differentiation but does not address the curse of dimensionality. For high-dimensional PDEs, the cost of maintaining and computing spectral coefficients could be prohibitive. The paper does not discuss how the method scales beyond low-dimensional problems.

**Essential References Not Discussed:**

The paper does not cite relevant works on neural networks with spectral methods, including:
- Fanaskov et al., 2023 – Spectral Neural Operators (Doklady Mathematics, 2023)
- Choi et al., 2024 – Spectral Operator Learning for Parametric PDEs Without Data Reliance (CMAME, 2024)

**Experimental Designs Or Analyses:**

The method is evaluated on problems where spectral methods are expected to perform well. There are no experiments demonstrating performance on hyperbolic PDEs, non-smooth solutions, or high-dimensional problems where spectral methods typically struggle. Also, there is no discussion on how computational cost scales with problem dimensionality. Given that spectral methods can become expensive in high dimensions, a cost-benefit analysis comparing against other operator learning methods would be beneficial.

**Methods And Evaluation Criteria:**

The paper presents a reasonable experimental setup, but there are significant gaps in the evaluation:

- The paper evaluates common PDEs but does not test its robustness on problems known to be challenging for spectral methods, such as high-frequency Helmholtz equations or highly nonlinear PDEs. These cases would provide a better understanding of the method’s limitations.

- The paper does not clearly compute the costs associated with FFT, nor does it discuss how the complex number operations arising from first-order differentiation are handled efficiently.

**Other Comments Or Suggestions:**

N/A

**Other Strengths And Weaknesses:**

**Strengths:**

- The idea of leveraging spectral methods in neural networks is well-motivated and could be useful for problems where spectral methods naturally excel.
- The proposed approach effectively reduces memory consumption for automatic differentiation, making it a viable alternative for solving PDEs with high-order derivatives.

**Weaknesses:**

- The method does not extend well to irregular geometries, significantly limiting its practical applicability.
- Scalability is unclear, especially for high-dimensional problems.
- The method is likely ineffective for hyperbolic PDEs or non-smooth solutions, but no experiments are provided to evaluate these cases.
- The theoretical claims lack rigor, particularly regarding spectral convergence and sampling error analysis.

**Questions For Authors:**

- The proposed method does not support irregular geometries, which is a major advantage of PINNs. How do you plan to extend it beyond structured domains?
- The computational cost for high-dimensional PDEs could be expensive. Have you tested how well SINNs scale beyond low-dimensional problems?
- Given the reliance on spectral methods, how does the approach handle hyperbolic equations and non-smooth solutions? Have you conducted any experiments to evaluate performance on such cases?
- The convergence analysis does not consider sampling error. How do you justify the claim of spectral convergence without accounting for this factor?

**Relation To Broader Scientific Literature:**

The methodology of solving PDEs using machine learning aligns with the broader field of Scientific Machine Learning (SciML), which has shown significant potential in advancing computational efficiency and flexibility across various applications.

**Theoretical Claims:**

The convergence analysis does not take into account the impact of sampling errors, which could significantly affect the claimed spectral accuracy. In addition, the paper does not provide a theoretical discussion on the significant discrepancy between the experimental results and spectral convergence.

---

> ### Author Rebuttal · Authors · 2025-03-31
>
> Thanks for the positive assessment and thoughtful questions.
> # Challenging Problems
> Firstly, we consider a well-defined 1D Helmholtz equation as you suggested:
> $$
>  u_{xx} + \lambda^2u=0, x\in [0,2\pi],
> $$
> with the boundary condition
> $$
> u(0)=u(2\pi)=1, \quad u_x(0)=u_x(2\pi)=\lambda.
> $$
>
> The solution is $u=\cos(\lambda  x)+\sin( \lambda x)$. The results are as follows:
>
> | $\lambda$ | PINNs | SINNs |
> | --- | --- | --- |
> | 2 | $5.37\times10^{-4} \pm 5.30\times10^{-4}$ | $4.72\times10^{-6} \pm 3.39\times10^{-6}$ |
> | 4 | $9.02\times10^{-4} \pm 4.54\times10^{-4}$ | $1.90\times10^{-5} \pm 3.71\times10^{-6}$ |
> | 8 | $1.75\times10^{-3} \pm 6.41\times10^{-4}$ | $1.45\times10^{-4} \pm 1.87\times10^{-4}$ |
> | 16 | $4.80\times10^{-1} \pm 3.38\times10^{-1}$ | $8.63\times10^{-6} \pm 6.24\times10^{-3}$ |
> | 32 | $9.94\times10^{-1} \pm 7.57\times10^{-3}$ | $1.47\times10^{-3} \pm 1.46\times10^{-3}$ |
>
> For a 2D Helmholtz equation:
>
> $$
> \Delta  u+ 8u=0, \boldsymbol{x}\in \Omega = [0,2\pi]^2,
> $$
> with the condition:
> $$
> u(\boldsymbol{x})=g(\boldsymbol{x}), \boldsymbol{x} \in \partial \Omega,
> $$
> $$
> u_x(\boldsymbol{x})=p(\boldsymbol{x}), \boldsymbol{x} \in \lbrace 0 \rbrace \times[0,2\pi],
> $$
> $$
> u_y(\boldsymbol{x})=q(\boldsymbol{x}), \boldsymbol{x} \in [0,2\pi] \times \lbrace 0 \rbrace.
> $$
> $g,p,q$ are derived by $u=(\cos(2x)+\sin(2x))*(\cos(2y)+\sin(2y))$.
>
> The results are:
>
>  | PINNs | SINNs |
>  | --- | --- |
>  | $3.11\times10^{-3} \pm 1.97\times10^{-3}$ | $2.33\times10^{-4} \pm 7.31\times10^{-5}$ |
>
> Furthermore, we don't think this problem is a challenge for SINNs. Because solving the boundary-value problems are like optimizing:
> $$
> \min_{x} |Ax-b|^2.
> $$
> Thus, changing $\lambda$ is merely changing the location of the non-zero element in $x$ in SINNs.
>
> # Irregular geometries
> Due to the limitations, please read it in reviewer PNMU.
> # High-dimensional problems
> Thanks for pointing out this limitation. There is a difference between using SINNs to solve high-dimensional PDEs and low-dimensional PDEs: PINNs can handle high-dimensional problems because their error is dependent on $N$ which is the number of sampled points and independent of the dimensionality $d$. As SINNs sample $N$ frequencies, if the error is also independent of the dimensionality $d$, SINNs can address the curse of dimensionality (CoD). Fortunately, by utilizing the optimized hyperbolic cross,  (Shen et al., 2011) proved that the error is also independent of the dimensionality $d$ (Key result is Corollary 8.3). Some examples can be found in (Shen \& Yu, 2010).  Herein, we will apply high-dim Fourier operators by a sparse matrix to avoid CoD.
>
> And we show the scaling by an ideal example:
> $$
> \Delta u =f, \boldsymbol{x} \in [0,2\pi]^d
> $$
> with $u=\sum_{i=1}^d \cos(x_i)$ and $f=-\sum_{i=1}^d \cos(x_i)$. Here are the results  (memory and rate are for SINN):
> |$d$|PINN|SINN|Memory (MB)| Iteration rate (ite/s)|
> |--|------|-----|----|----|
> |2|$7.72\times 10^{-4}$|$5.34\times 10^{-4}$|542|992|
> |5|$5.09\times 10^{-3}$|$2.75\times 10^{-3}$|550|1000|
> |10|$1.24\times 10^{-1}$|$1.18\times 10^{-1}$|550|1000|
> |20|$7.25\times 10^{-1}$|$1.50\times 10^{-1}$|790|982|
> |100|$1.04\times 10^{0}$|$1.27\times 10^{-1}$|8736|995|
>
> # Theoretical Claims
> The sampling error exists in PINNs because the Monte Carlo method is used to estimate the integration. For example, the residual term:
> $$
> \min_{\theta} \int_{\Omega}|\mathcal{L}_r(x;\theta)|^2 d x \approx \min _{\theta}\mathcal{L}= \min _{\theta} \sum _{i=0}^M \left|\mathcal{L}_r\left(x_i;\theta\right)\right|^2
> $$
> As the integration domain contains infinite points, using finite points to estimate it always has an error, i.e. the sampling error in Monte Carlo methods. However,SINNs don't need Monte Carlo methods because we don't have integrations and our input domain is finite: the frequencies. Herein, our loss function is directly:
>
> $$
>     \min_{\theta} \tilde{\mathcal{L}}= \min _{\theta} \sum _{i=0}^M|\tilde{\mathcal{L}}_r(k_i;\theta)|^2
> $$
>
> With the assumption that the network is powerful enough (i.e. regardless of the sampling method selected, finally, the network learns the total dataset). Then $\tilde{\mathcal{L}}=0$ means the output of SINNs exactly equals to the coefficients of the frequencies.  But  $\mathcal{L}=0$  only means the output of PINNs is accurate at the points  $ \lbrace x_i \rbrace_{i=1}^M$ but still has the sampling error from Monte Carlo methods.
>
> In conclusion, because we truncate the Fourier series to a finite number, SINNs have the spectral error but don't have the sampling error.
>
> # Essential References
>
> Thanks for the given relevant papers. However, the frameworks of PINNs and Neural Operators are different. PINNs aim to learn the map from a definition domain to a functional space while NOs learn the map from a functional space to a functional space. Herein, we didn't write the review of Neural Operators in our manuscript. But, indeed, we should write a brief review of neural networks with spectral methods in other fields.

---

> > ### Comment · Reviewer_zt1d · 2025-04-05
> >
> > The paper's experiments primarily focus on PDEs that are easily solvable using spectral methods and omit more challenging cases, such as hyperbolic PDEs. Furthermore, the theoretical claims provided address only the sampling error, whereas my original concern pertained specifically to spectral convergence. Additionally, the selected baseline—a plain PINN method from seven or eight years ago—is somewhat outdated, and the paper lacks comparisons with more recent methods specifically developed to address high-frequency or spectral bias issues (e.g., PINNs enhanced by positional encoding). Given these limitations, I will maintain my current score.

---

> > > ### Author Response · Authors · 2025-04-05
> > >
> > > Thanks for your further discussion of our paper.
> > >
> > > # Omit more challenging cases
> > >
> > > We apologize for having wrongly thought that solving the Helmholtz equation that you mentioned could address your concern.
> > >
> > > Here we choose the classical hyperbolic PDEs: wave equations. We consider the following wave equation from PINNacle (arXiv:2306.08827):
> > > $$
> > > u_{tt}-4u_{xx}=0, \quad x\in [0,2\pi], t \in [0,1]
> > > $$
> > >
> > > $$
> > > u(x,0)=\sin(x)+ \sin(4 x) / 2, \quad x\in [0,2\pi].
> > > $$
> > >
> > > $$
> > > u_t(x,0)=0, \quad x\in [0,2\pi].
> > > $$
> > >
> > > with the solution $u=\sin(x) \cos(2t) + \sin(4  x) \cos(8t) / 2$
> > >
> > > The results are as follows:
> > > |PINN|SINN|
> > > |--|--|
> > > |$1.49\times 10^{-1}$|$3.15\times 10^{-5}$|
> > >
> > > You can find other methods from Table 3 Wave1d-C in PINNacle. The best one is $9.79\times 10^{-2}$ which uses NTK loss reweighting.
> > >
> > > Hope the results can address your concern. If not, please give us a concrete equation.
> > >
> > > # The convergence analysis does not take into account the impact of sampling errors, which could significantly affect the claimed spectral accuracy.
> > >
> > > As SINNs don't have sampling errors, they will not influence the spectral accuracy theoretically. And we admit that it is hard to discuss the discrepancy between the experimental results and spectral convergence.
> > >
> > > # More recent methods specifically developed to address high-frequency
> > >
> > > Our baselines not only have the vanilla PINNs but also include VSPINN (Hoon Song et al., 2024)  which is accepted by NeurIPS2024.
> > >
> > > As for the positional encoding, we already have the experiments of Fourier Embedding (FE) which is a representative method in PINNs. As we stated, those methods are also valid for our SINNs, so we only did experiments on SINN combined with FE. We can understand your concern, so we did experiments to compare SINN+FE and PINN+FE with different embedding channels $N_{FE}$ ($N_{FE}=0$ means without FE) under the same setting as in Section 4.5:
> > >
> > > |$N_{FE}$|0|2|4|8|
> > > |--|--|--|--|--|
> > > |PINN+FE|$3.73\times 10^{-3}$|$2.11\times 10^{-3}$|$1.79\times 10^{-3}$|$2.48\times 10^{-3}$|
> > > |SINN+FE|$3.17\times 10^{-3}$|$3.31\times 10^{-4}$|$1.74\times 10^{-3}$|$3.34\times 10^{-3}$|
> > >
> > > We hope the above supplementary can have a positive impact on your evaluation。

---

### Decision · Program_Chairs · 2025-05-01

**Decision:**

Reject

**Comment:**

The paper uses spectral derivatives in place of derivatives computed through AD in physics informed neural networks (that enforce PDE constraints as soft penalties in the loss) and reformulate the problem in spectral space. The reason is to deal with the high memory/compute cost of higher order derivatives for AD. Experiments demonstrate this advantage clearly. Further, their method retains spectral convergence properties.

Strengths identified:
* Simple idea from classical methods to deal with current bottlenecks in PINN training. All reviewers acknowledged that this can be very useful in certain settings. Experiments demonstrate this clearly.

Weaknesses:
* Spectral derivatives are well-known in classical methods - for the PDEs evaluated it seems spectral methods would work well in the classical sense (with pseudospectral solvers etc) - it's unclear if the experiments were chosen for these specific kind of problems; whereas PINNs are more general and mesh agnostic.
* FFT based methods bring standard issues that reviewers raised - aliasing, conforming to regular geometries and uniform grids. While the authors had good arguments for all (including methods from classical techniques like fictitious domain method for irregular geometries, for example), the experiments were minimal to demonstrate this in the rebuttal.

Overall, the reviewers acknowledged that the contributions were positive and useful. The paper could be made better with more emphasis on harder PDEs (where spectral methods fail in the classical sense due to heavy aliasing etc; solutions with discontinuities visualized), with examples on complex geometries and unstructured grids to show that the FFT based differentiation is still performant and useful compared to standard variants of PINNs. While the authors attempted to answer this during the rebuttal, the experiments were minimal (due to the short response period) with simple examples chosen.